# Quantifying task-relevant representational similarity using decision variable correlation

**Yu (Eric) Qian**
Department of Neuroscience
The University of Texas at Austin
`ericqian@utexas.edu`

**Wilson S. Geisler**
Department of Psychology
The University of Texas at Austin
`w.geisler@utexas.edu`

**Xue-Xin Wei**
Department of Neuroscience
The University of Texas at Austin
`weixx@utexas.edu`

## Abstract

Previous studies have compared neural activities in the visual cortex to representations in deep neural networks trained on image classification. Interestingly, while some suggest that their representations are highly similar, others argued the opposite. Here, we propose a new approach to characterize the similarity of the decision strategies of two observers (models or brains) using decision variable correlation (DVC). DVC quantifies the image-by-image correlation between the decoded decisions based on the internal neural representations in a classification task. Thus, it can capture task-relevant information rather than general representational alignment. We evaluate DVC using monkey V4/IT recordings and network models trained on image classification tasks. We find that model–model similarity is comparable to monkey-monkey similarity, whereas model–monkey similarity is consistently lower. Strikingly, DVC decreases with increasing network performance on ImageNet-1k. Adversarial training does not improve model–monkey similarity in task-relevant dimensions assessed using DVC, although it markedly increases the model–model similarity. Similarly, pre-training on larger datasets does not improve model–monkey similarity. These results suggest a divergence between the task-relevant representations in monkey V4/IT and those learned by models trained on image classification tasks.

## 1 Introduction

Deep learning [1; 2] has substantially impacted how neuroscientists construct brain models. For vision neuroscience, deep neural networks offer candidate models for studying the primate ventral pathway [3; 4; 5] and, more recently, dorsal pathway [6; 7; 8]. Early work reported that the representations in convolutional neural networks (CNNs) trained on image categorization tasks can explain a substantial fraction of variance in high-level visual areas, surpassing classic models for these areas [3]. Follow-up research has tested many variants of deep networks on their alignment with the brain using both neural data [9; 10; 11] and behavior data [12; 13; 14]. One appealing hypothesis is that deep networks that exhibit higher accuracy and robustness in vision tasks, or trained on larger datasets would better explain visual processing in the brain.

One important question is how to compare deep network models and the brain. One class of methods seeks to quantify the similarity of internal representations between models and brains. This includes methods such as representational similarity analysis (RSA) [15], linear regression [5; 16], and generalized shape metrics [17; 18]. More recently, another class of methods that put more emphasis

39th Conference on Neural Information Processing Systems (NeurIPS 2025).

on quantifying the behavioral similarity has been proposed, including Cohen's Kappa [13] and I2n behavioral predictivity [12; 19]. The goal of these methods is to provide an image-by-image comparison of the decision strategies used by neural networks and the brain. One challenge has been how to properly disentangle the model accuracy, decision biases and decision consistency from behavior [20; 13]. Intriguingly, studies seem to find contradictory trends in model-brain alignment. While some studies suggest that deep learning systems converge to learning a common representation [21; 22], others suggest that the similarity will not grow indefinitely or has capped [23; 24]. The reason for this remains unclear.

Here, we propose a principled approach that combines the merits of model comparisons at the representation and behavior levels. Our method is based on decision variable correlation (DVC) developed to measure the behavioral similarity based on choice data [20] in signal detection theory, and we have generalized it to analyze the consistency of high-dimensional neural representations. While prior work developed techniques to estimate DVCs from behavioral data [20], we instead infer DVCs from neural representations. Thus, our proposed DVC metric should be interpreted as a measure of the representational similarity. Our approach specifically quantifies the trial-by-trial consistency of two neural representations for solving a classification task, ignoring features that are irrelevant for the task. In doing so, our approach enables one to infer the consistency of the decision strategy of two observers from their internal representations based on the assumption of optimal linear readouts.

Applying our method to compare neural representations for solving image recognition tasks from monkey brains and deep network models led to several interesting findings. In particular, we found that model–model and monkey–monkey similarities are comparable, whereas model–monkey similarity is consistently lower and decreases with increasing ImageNet-1k accuracy. Somewhat surprisingly, this gap is not remedied by adversarial training or training on larger datasets.

## 2 Background and relevant work

Community efforts have pushed towards better methods to compare brains and models and for brain-model alignment. Different factors have been hypothesized to be relevant for alignment, including model architecture, robustness, and training data, as summarized below.

**Model architecture and scale** One hypothesis has been that as models improve in task performance or architectural complexity, their internal representations become more brain-like. The Brain-Scores [19] of the image classification models were reported to be positively correlated to ImageNet-1k accuracy, although the trend plateaus at higher accuracy. On the other hand, studies using RSA reported that neither model scale nor architecture significantly improved alignment to human behavioral similarity judgments [9]. Another study using RSA reported negative correlation between alignment to human neural activity and model complexity [25]. Studies using Cohen's Kappa reported that human-model behavioral consistency at the image level remains low despite improved performance on out-of-distribution datasets with scaling[13; 14].

**Robustness** The primate visual system is robust against external and internal noise, prompting the question of whether robustness to adversarial perturbations or corruptions is related to brain-model alignment [26]. Recent work proposed that adversarial robustness might promote the learning of representations better aligned with human perception [27]. By enforcing alignment with monkey IT representations, models exhibited both enhanced adversarial robustness and increased behavioral alignment with human subjects [26]. Another study found that model metamers – artificial stimuli that elicit the same response as natural stimuli, generated by robust models– are more recognizable to humans, but are not themselves predictive of recognizability [28]. However, studies using Cohen's Kappa report that robust models still diverge from humans in their error patterns[14]. Additionally, there has also been evidence suggesting that even though adversarial training increased model robustness, these robust networks may not use human-like features unless explicitly aligned [29; 30].

**Rich and multimodal training data** Using Cohen's Kappa, [14] reports that models trained on larger and more diverse datasets become more human-like in their behaviors. On the other hand, a recent large-scale study using a variation of RSA reported that upgrading from ImageNet-1k to ImageNet21k does not significantly improve alignment to human brain, but object-oriented ImageNet datasets lead to much better alignment than datasets containing only places or faces [11]. Similarly, an ecologically-motivated dataset seems to improve model-brain alignment [31]. Joint vision-language models such as CLIP have also been shown to better predict human brain activity [14; 10; 32].

**Similarity measures** Classic methods such as linear predictivity measure how well neural responses can be predicted from network representation [3]. Representational similarity analysis (RSA) [15; 4] is a popular approach that quantifies the similarity in the geometrical structure of two representations, and is blind to the specific features used to solve a task [33]. Other shape metrics such as centered kernel alignment (CKA) and canonical correlation analysis (CCA) also do not specifically address the similarity in task-relevant features. Measures of behavioral similarity such as error consistency are task-focused [13], yet they may be overcalibrated on accuracies of the observers and are therefore sensitive to the choice of decoders[13; 14] as we will show later. Recent studies highlighted the challenges in the interpretation of results based on these methods [19; 11], *e.g.*, different methods for quantifying the brain-model similarity could lead to different conclusions [34; 35]. As the field continues to develop methods for analyzing and interpreting model-brain alignments [36; 37], it would be desirable to have principled, task-relevant, accuracy-agnoistic measures to better illustrate the possible divergence between brains and models.

## 3 DVC: Quantifying the trial-by-trial consistency of two representations

We develop a new method to evaluate the consistency of two neural representations. This method is based on a principled generalization of signal detection theory. It enables one to estimate how correlated the decision strategies of two observers are for a classification task. The method is robust to the observers' biases and is not confounded by the behavioral accuracy. It operates at the level of neural representations, and enables one to analyze the internal representation to infer the consistency of the two representations for solving the classification tasks. Thus, the method can quantify task-relevant representational similarity. Compared to methods purely based on behavior [20; 13], it takes advantage of the richness of the internal representations of neural networks and brains. Meanwhile, in contrast to methods for analyzing the similarity of two neural representations (such as representational similarity analysis), our method focuses on the dimensions that are relevant for a behavioral task and is invariant to variability along other task-irrelevant dimensions.

### 3.1 Decision variable correlations (DVC) of two neural representations

Signal detection theory is fundamental in the study of perceptual behavior. The idea is that, for binary-choice tasks, observer uses a continuous decision variable (DV) to make a choice (Fig. 1a). Recently, [20] proposed to generalize signal detection theory to study the correlation of decision variables of two observers (Fig. 1b). Their method inferred the DVC from binary choice data. Here, we develop a simple new strategy to infer trial-by-trial DVCs from high-dimensional internal representations (Fig. 1c).

For a pair of image categories and an observer (a brain area or a particular layer from a neural network), we can take its neural representation and find the optimal decision axis for solving the categorization task. We then project the high-dimensional representation for each image onto the decision axis and obtain its decision variable. Now consider the case of two observers. By performing the analysis on both observers, we obtain two decision variables for each image. We can compute the correlation of the decision variables (DVC) for the two observers (Fig. 1c). This correlation captures the similarity of the encoding and the decoding into a decision for the two observers in this classification task.

Note that the method of inferring DVC from behavioral responses only applies to binary choice tasks. Our new method does not suffer this limitation. Given N (>2) image classes, we can focus on each pair of categories at a time, and infer the DVC for that particular classification task.

### 3.2 Implementation of the DVC method

We next discuss how we implement the DVC framework to analyze the high-dimensional neural representations. The code is available at `https://github.com/wei-bbc-lab/DVC`.

**Decoding decision variables (DVs) from neural representations** For each pair of classes (e.g., cats v.s. dogs), we use Linear Discriminant Analysis (LDA) to find the axis that maximizes class separation to decode the DVs from the brain or model representations. The projection onto the LDA axis reflects the model's tendency to classify the image as one class versus the other; values near the midpoint indicate greater classification uncertainty. One important practical issue is that LDA can be unstable under high dimensions with few samples. The reason is that there are many noisy feature directions with similar class separation, but the projections of image representations along these dimensions can be different. Consequently, even if two models have the same underlying

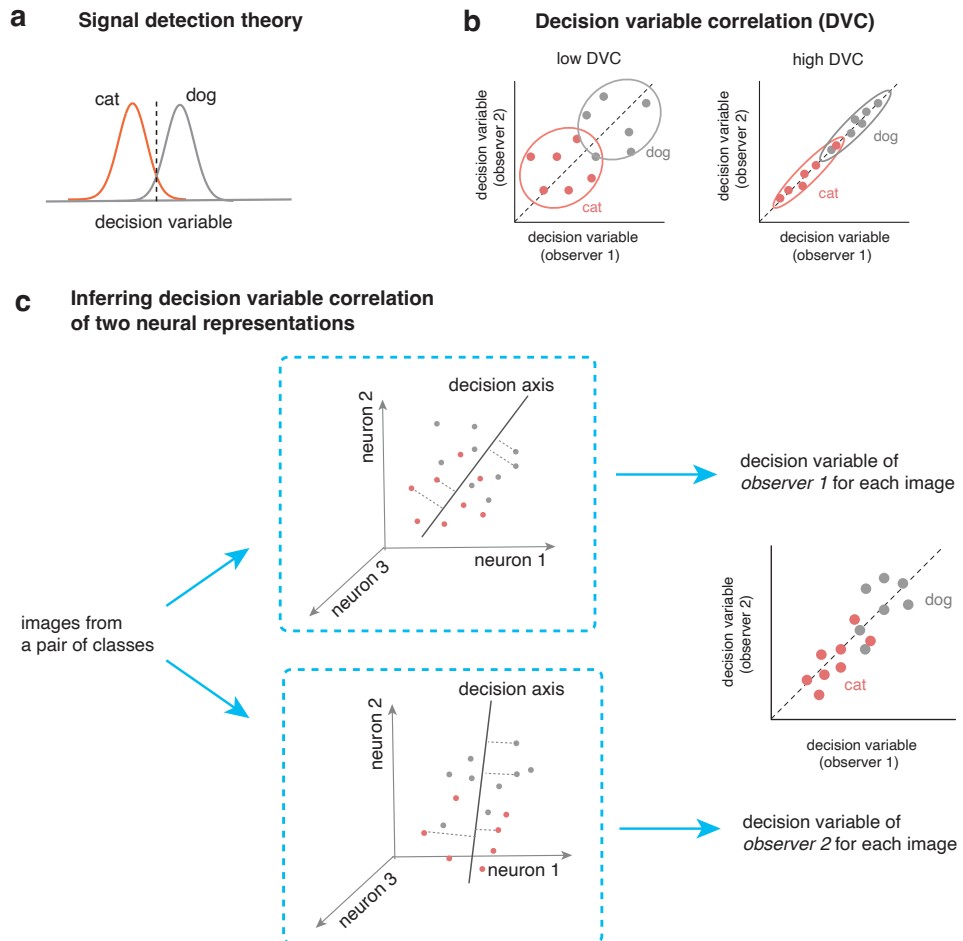

Figure 1: **The computational framework of decision variable correlation (DVC) for neural representations**. (a) Traditional signal detection theory models how a single observer solve a binary classification task. The idea is that the observer use a decision variable together with a criterion (dash line) to make a choice. (b) Decision variable correlation generalizes the signal detection theory to study the trial-by-trial consistency of the decision variables of two observers. The two panels illustrate two cases with the same accuracy in solving the task, but with drastically different correlations in the decision variabless (DVs). (c) We further generalize DVC to compare two neural representations. The basic idea is to use optimal linear classifier to infer the decision variables of individual observers and then quantify the consistency of the decision variables.

representations, LDA projections may show low correlation. Note that models examined in this paper have a wide range of dimensionality in their penultimate layer ($10^3 - 10^7$).

To address this problem, we use dimensionality reduction (e.g., PCA) to reduce the representations to the same number of features before using LDA to decode the underlying DV [1]. We measure the similarity between decoded DVs using Pearson Correlation. A DVC value is obtained for each class in each pair of classes. The final reported number is taken as the average of all DVC values.

**Normalization to address measurement noise in DV** Measurement noise may limit the accuracy of the inferred DVC. The otherwise perfect correlation between two identical representations would be corrupted by adding noise. Low correlation might therefore reflect true underlying dissimilarity or high noise level. To correct for the under-estimation of DVCs due to measurement noise, we develop a split-half procedure to infer the impact of noise.

We aim to estimate the true correlation between two decision variable (DV) signals, $DV_A$ and $DV_B$, each of which is contaminated by independent noise. To correct for the attenuation bias introduced

---

[1]25 PC dimensions. See Appendix C.3 and C.4 for experiments that demonstrate the robustness of the results.

by noise, we split each DV into two independent halves: $DV_{A1}$, $DV_{A2}$ and $DV_{B1}$, $DV_{B2}$. For neural recordings, this would indicate splitting into two sets of neurons, and for model representations, two sets of hidden units. We then compute a noise-corrected Pearson correlation as follows:

$$\rho_{\text{corrected}} = \frac{r_{\text{cross}}}{r_{\text{self}}} \tag{1}$$

where the numerator reflects the geometric mean of all pairwise cross-observer correlations:

$$r_{\text{cross}} = \left[ \rho(DV_{A1}, DV_{B1}) \cdot \rho(DV_{A1}, DV_{B2}) \cdot \rho(DV_{A2}, DV_{B1}) \cdot \rho(DV_{A2}, DV_{B2}) \right]^{1/4} \tag{2}$$

and the denominator normalizes by the geometric mean of the within-observer (split-half) reliabilities:

$$r_{\text{self}} = \left[ \rho(DV_{A1}, DV_{A2}) \cdot \rho(DV_{B1}, DV_{B2}) \right]^{1/2} \tag{3}$$

This correction removes the bias introduced by independent, additive symmetric noise in the estimated decision variable, yielding an unbiased estimate of the true underlying correlation between the latent signals driving $DV_A$ and $DV_B$ [2]. How to best recover DVC under more general noise conditions remains an interesting question for future work.

## 4 DVCs reveal the divergence between deep networks and brains

We apply the new DVC-based methodology to examine (i) the trial-by-trial consistency of the neural representation of the high-level visual areas (V4/IT) between macaque brains, (ii) the consistency between individual neural network models & the IT/V4 neural representations, and (iii) the consistency between different deep neural network models. We specifically consider three classes of deep network models: (i) models that were pre-trained on ImageNet-1k using standard network training (i.e., no adversarial training); (ii) "robust models" that were fine-tuned on ImageNet-1k using adversarial training; (iii) "data-rich models" that were pre-trained on even larger datasets such as ImageNet-21k and JFT-300M.

### 4.1 High trial-by-trial consistency of V4 & IT representations across monkey brains

We evaluate the consistency of neural representations in different macaque monkeys. We used the publicly available dataset of objects rendered on naturalistic scenes [38]. In these experiments, they used images from eight classes {animals, boats, cars, chairs, faces, fruits, planes, tables}, with 400 images each, totaling $400 \times 8 = 3200$ images. Recordings were taken from V4 and IT areas of two adult macaque monkeys passively viewing these images. The brain representation is taken to be the time-binned spike counts averaged over 50 repeats. 100 neurons from V4 and IT combined were obtained from each monkey.

We first examine the classification accuracy based on V4 and IT response, and find that the accuracy based on LDA is high (0.94 and 0.92, respectively). We combine the neural data from areas V4 and IT, and compute the DVCs. We find that the DVC between the monkeys is about 0.57. We further compute the DVCs for V4 and IT separately, and find the DVC values to be 0.63 and 0.41, respectively. Overall, these results suggest that DVCs across the monkeys' brain are generally high, implying that the encoding and the decision strategies used by different monkeys are consistent on an image-by-image basis.

### 4.2 Deep networks with higher accuracy on ImageNet exhibit lower DVCs with brains

We study a set of models (n=43, obtained from Torchvision, [39]) pretrained on ImageNet-1k, an influential benchmark in image classification. This also offers a fair comparison between models by controlling for confounding factors related to different training data. We use the same neural datasets from [38] as above. We feed the images in [38] into deep vision-based neural networks, subject to standard transforms. The model representation for an image is defined as the activation taken from the penultimate layer – the last layer before the final logit layer.

**Brain vs. network** Evaluating the DVCs between models and monkey brains, we find that the consistency between models and monkeys are modest, and generally lower than that of monkeys. For the 43 models we tested, the average is $0.29 \pm 0.05$. Given the differences in the training data,

---

[2]See Appendix A for proof, and Appendix C for discussions on the validity of DVC under different conditions, including behaviors of split normalization at boundary conditions.

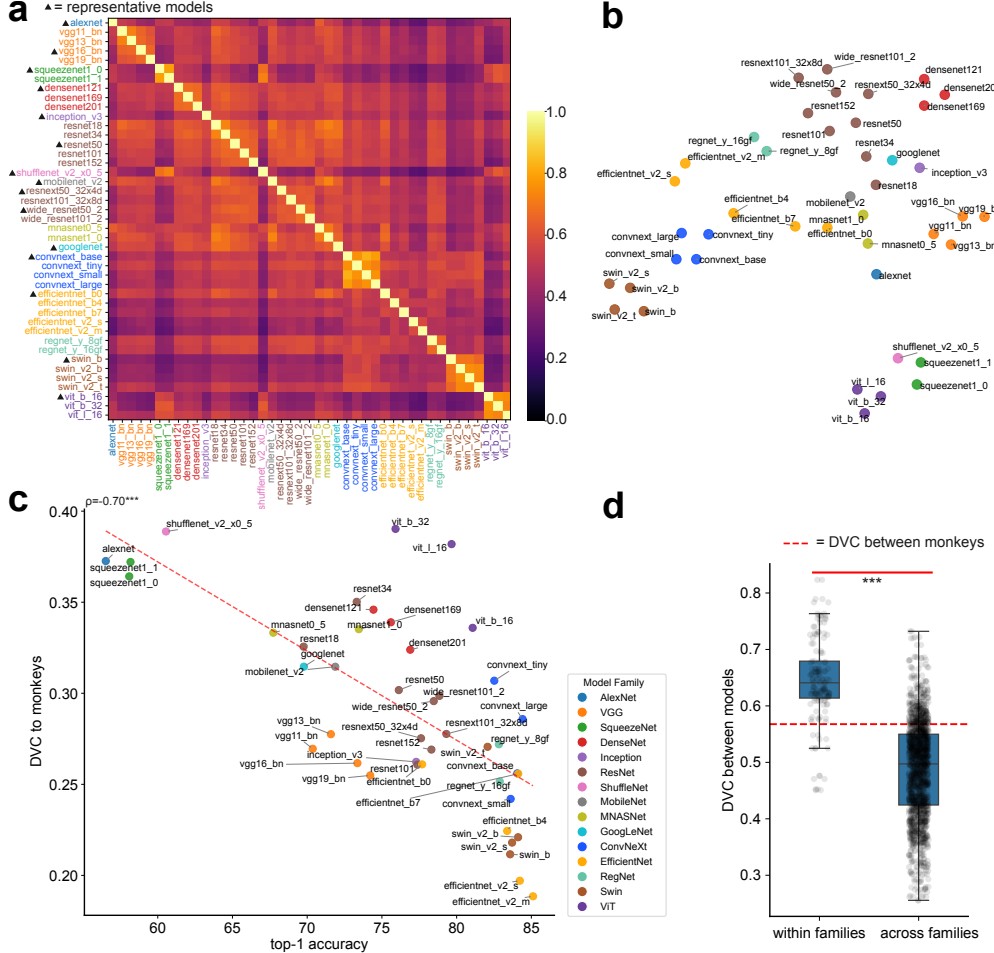

Figure 2: **Results on models trained on ImageNet-1k**. (a) Heatmap: DVCs inferred for pairs of models. Different colors are used to indicate models from different model families. 15 models are selected to represent this cohort in later analysis. (b) 2D t-SNE embedding of the models using their dissimilarities, measured as $1 - DVC$. (c) There is a strong *negative correlation* between the classification performance (top-1 accuracy) of a network and its DVC correlation to the V4/IT representation. (d) Networks belonging to the same family exhibit higher DVCs compared to those belonging to different model families (p = 1.33e-56).

learning algorithm, and loss functions between deep networks and brains, this is perhaps not too surprising. The models we tested differ in their ability to solve image categorization tasks. One influential hypothesis has been that the more accurate a network can solve the task, the more similar its representation would be when compared to that of primate visual cortex. Earlier results [19] supported this hypothesis. This motivated us to examine whether networks with higher performance on ImageNet-1k also have higher DVC with macaque IT/V4. Surprisingly, we find the opposite, that is, networks with higher top-1 accuracy on ImageNet-1k generally have lower DVC with IT/V4 representation (Pearson correlation = -0.70, p = 2.28e-07; Fig. 2c).

**Network vs. network** We next examine the DVCs between different deep neural networks. Specifically, we evaluate DVCs between deep networks from different model families[3]. Using DVC, models from the same family or otherwise share architectural similarities are clearly clustered together(Fig. 2a,b). We find that DVCs between pairs of models within the same family (similar model structures and training processes) are substantially higher than pairs from different families (p =

---

[3]We define a model family as a set of architectures sharing a canonical computational graph – such as residual, attention, or convolutional block structures—with variation limited to hyperparameters like depth, width, patch size, or token embedding dimension. See Appendix B for more model details

1.33e-56, Fig. 2d), consistent with previous findings [40]. We also find DVCs between models not to be exceedingly high. Despite being trained on the same dataset, they do not seem to converge to a single solution, at least not significantly higher compared to the similarity between the two monkeys (Fig. 2d). These results imply that model structures and training processes still play significant roles in the task solutions found by the models. Notably, these results differ substantially from results obtained by computing error consistency. These studies [13; 14] reported that (i) the consistency between model and brain is very low; (ii) the consistency between network models is much higher than the consistency between humans. Later, we will address the difference between the methodologies.

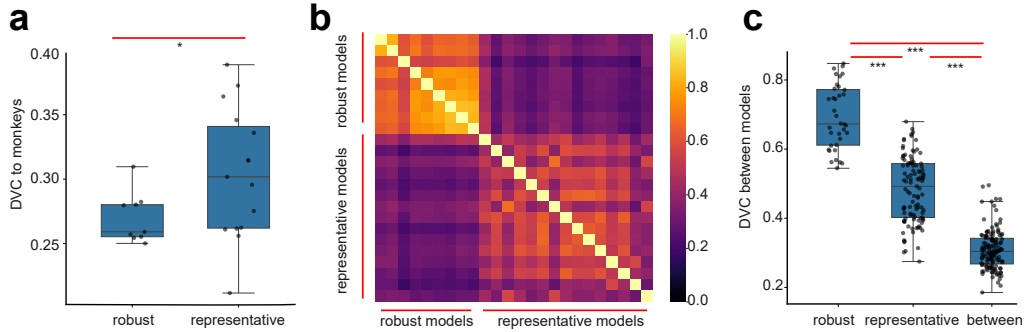

Figure 3: **Results on robustly trained deep networks on ImageNet-1k**. (a) Networks based on adversarial training has lower DVC with V4/IT compared to the representative models (introduced in Fig. 2)) without adversarial training. (b) Heatmap showing the inferred DVCs between pairs of models. (c) Robustness networks have high DVCs among themselves, and they have relatively low DVCs with the representative models.

### 4.3   Adversarially trained networks, while highly consistent, have low DVCs to the brain

Robustness represents one important difference between deep networks and our perceptual systems. Small perturbations to images that are imperceptible to humans can lead to qualitative errors in deep networks (i.e., adversarial examples) [41]. Adversarial examples reflect the misalignment between representations in deep networks and brains, given certain *local* perturbations in the inputs. Adversarial robustness can be increased by using adversarial training, e.g., by finding adversarial examples and adding them to the training set. Studies suggest that features learned through adversarial training may be more aligned with human perception [27; 28], posing an intriguing hypothesis that by making networks locally consistent with human perception, network representations may be better aligned with brain representations *globally*.

To test this hypothesis, we examine the DVCs of a set of adversarially trained networks and macaque V4/IT. We obtain robust models fine-tuned for adversarial robustness on ImageNet-1k(n=9, from Robustbench, [42]). Evaluating the DVCs of these models to V4/IT representations, we observe no improvement in the similarity to the brain. In fact, we observe a slight decrease of the DVC values $(0.27 \pm 0.02$, Fig. 3a). Furthermore, we observe that models based on similar adversarial training procedures show a high similarity with each other $(0.69 \pm 0.09$, Fig. 3c). Meanwhile, their similarities to models without adversarial robustness drop substantially (p = 5.203e-37, Fig. 3c). These results suggest that adversarially trained models converge to a common solution (despite that these models have different architectures). Their representations diverge from the non-adversarially trained deep networks, yet they also diverge from the neural representation in macaque V4/IT.

### 4.4   Networks trained on rich datasets exhibit no increase in DVC to the brain

Whereas ImageNet-1k has been an important benchmark dataset for the image classification community for a decade, recent state-of-the-art models are trained on larger datasets such as ImageNet-21k, which is a scaled-up version of ImageNet-1k, and JFT-300M, which is proprietary. Models trained on larger, more diverse datasets may generalize within a larger domain, and may show better out-of-distribution generalization ability. A recent study showed that models trained on these larger datasets may exhibit better alignment with human behavior [14]. Furthermore, the negative correlations between classification performance and DVCs to the brains (Fig. 2c) suggest the possibility of overfitting to a particular dataset that is much smaller than what brains are trained on evolutionarily,

developmentally, and during the experiments [43]. Therefore, it is of particular interest to investigate whether models trained on the richer datasets exhibit higher DVCs to the brain.

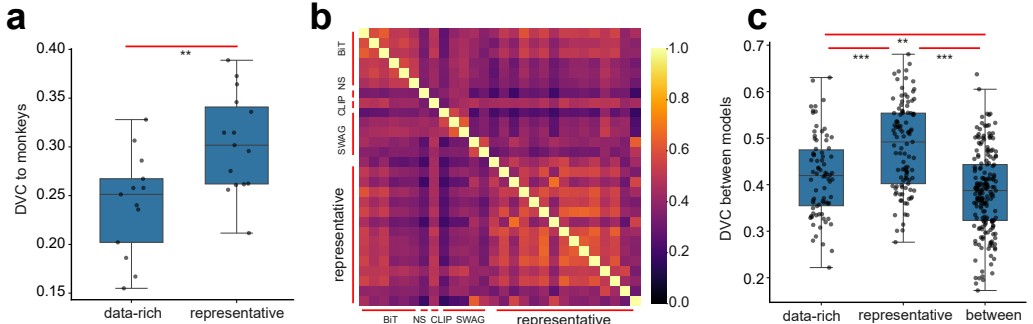

Figure 4: **Results on deep networks trained on richer datasets**. (a) Networks we examined that were pre-trained on richer datasets exhibit lower DVC with V4/IT compared to the representative models (trained on ImageNet-1k). (b) Heatmap showing the inferred DVCs between pairs of models. (c) DVCs between lower data-rich models and representative models are generally lower than those within representative models or data-rich models.

We examine models pre-trained on bigger, or multimodal datasets (n=13), namely, 5 SWAG models [44] from Torchvision and 6 BiT (Big Transfer)[45] models, Noisy Student[46] and CLIP [47] from Timm (for details, see Appendix B) [39; 48]. Briefly, BiT (Big Transfer) is a supervised pretraining approach that trains ResNetV2 models on large-scale datasets like ImageNet-21k. Noisy Student is a semi-supervised learning framework that iteratively trains a student model on both labeled (ImageNet-1k) and unlabeled (JFM-300M) data using noise-augmented inputs. CLIP is a contrastive vision-language model that jointly learns aligned image and text embeddings from web-scale paired data. SWAG is a training strategy introduced by Meta that improves supervised learning by using large-scale weak supervision from hashtAGs. All of these models enjoy better performance on ImageNet-1k than their vanilla counterparts. Comparing these models to V4/IT, surprisingly, we find that the DVCs to the brains are lower than those trained on ImageNet-1k ($0.24 \pm 0.05$, Fig. 4a). Given that these models generally have high ImageNet-1k accuracy, it seems to follow the previously reported trend that better performing models tend to show less consistency with brains. These data-rich models are less similar to the representative models trained on ImageNet-1k compared to the similarity among themselves (Fig. 4c).

## 4.5 Comparison to error consistency based on Cohen's Kappa

One method that is highly relevant to DVC is Cohen's Kappa. As a classic statistical measure of inter-rater consistency [50], Cohen's Kappa was recently applied to quantify the error consistency of deep networks and brains [13; 14]. These studies examined human-model alignment using categorical judgements on assorted out-of-distribution stimuli. For example, 'edges' where only the edges are illustrated, or 'silhouettes' which are filled outlines of objects. These studies arrived at very different conclusions, namely that (i) model-model similarity is significantly higher than human-human similarity, (ii) model-human similarity is extremely low and that (iii) models trained on rich datasets are more aligned with humans. At a high level, these results seem to be inconsistent with our findings, because we found that (i) deep networks exhibit modest consistency with the brain; (ii) DVCs between different deep networks trained on ImageNet-1k are not exceedingly large; (iii) models trained on rich datasets have lower DVCs with brains.

To understand these potential discrepancies, we performed several analyses. We start by applying Cohen's Kappa to study the neural dataset used above. We used 5-fold cross-validated logistic regression to obtain model decisions as well as monkey 'decisions'. Using this decoder, we find that DVC shows a high correlation with Cohen's Kappa, consistent with the theoretical analysis in [20] (Fig.5b,c). We find that Cohen's Kappa between deep networks and the brain is modest ($0.13 \pm 0.04$), and generally larger than the typical values reported in [13]. Furthermore, Cohen's Kappa between different deep networks ($0.23 \pm 0.07$) is not substantially larger than that between the monkeys (0.22). These results suggest that Cohen's Kappa applied to optimal linear classifiers leads to generally consistent results on this dataset. What then is causing the difference in network-

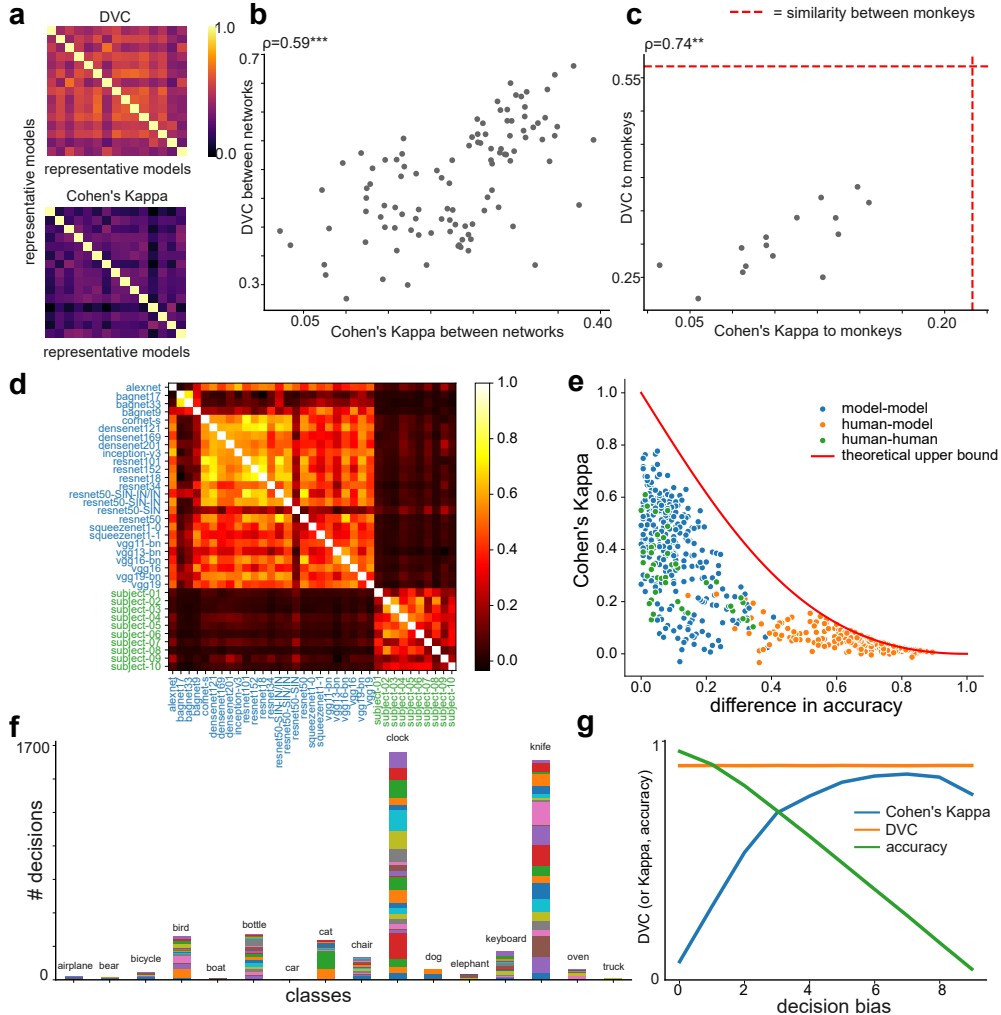

Figure 5: **Comparsion to Cohen's Kappa**. (a) Heatmaps showing the DVC and Cohen's Kappa for pairs of representative models. (b) There is a strong positive correlation between Cohen's Kappa and DVC on model-model consistency (evaluated on this dataset) (c) There is a decent positive correlation between Cohen's Kappa and DVC on model-monkey consistency. (d) The values of Cohen's Kappa between models (blue) and human subjects (green) are low while Cohen's Kappa between models and between human subjects are high, consistent with the original report. Re-analyzed based on data from [49]. (e) Scatter plot showing the relationship between Cohen's Kappa and the difference in accuracy of pairs of observers, and the theoretical upper bound. (f) The response histogram to 'edges' distortion based on the model and decision rules used in [13] and the original study [49]. Different colors represent different nerual networks. (g) Simulation results show that Cohen's Kappa is sensitive to decision biases, while DVC is invariant to decision biases.

network and network-brain consistency between the results reported in [13] and DVC? To address this question, we next analyze the behavioral data used in [13].

**Accuracy difference** While Cohen's Kappa was originally proposed to disentangle the accuracy and consistency of the observers under certain conditions, it remains possible that the two are intermingled in practice [13]. To assess this issue, we examine the behavioral responses to 'edges' stimuli from [49]. From the results shown in Fig. 5d,e, it is evident that Cohen's Kappa is strongly affected by the difference in the accuracy of the two observers[4]. When the difference in accuracy is high, Cohen's

---

[4]Theoretical upper-bound is given by $\kappa \leq \frac{(1-d)^2}{1+d^2}$, where $d$ is the accuracy difference between the two observers. See Appendix A.2 for the derivation.

kappa is low. This suggests that the low error consistency between humans and models reported in [13] is at least partially due to the large differences in accuracy between the two observers. It also potentially explains why models trained on 'rich' datasets, which enjoy better out-of-distribution performance, also exhibit better alignment with human behavior based on Cohen's Kappa [14].

**Decoder bias** According to signal detection theory, Cohen's Kappa is determined by both the correlation of the decision variables and decision criterion. Thus, we wondered if the extremely high Cohen's Kappa values between different networks as reported in [13] are due to biases in the decisions. We thus perform a second analysis to further investigate the data from [49] and analysis used in [13]. Consistent with our hypothesis, we find that the approach in [13] introduces high decision bias (see Fig. 5f) and reduced accuracy, especially when target categories do not align cleanly with the original training labels. We also find that the origin of this large decision bias is because the analysis in [13] is based on an aggregated decoder that estimates class probabilities by combining probabilities from related ImageNet-1k classes. Once we substituted the original decoder with a cross-validated logisitic regression classifier, the estimated Cohen's Kappa values become largely consistent with the DVC we obtained on the main dataset we analyzed. These results suggest that the large Cohen's Kappa values between different networks are due to the biases in the decoders[5].

These results highlight the advantages of the DVC method. DVC is insensitive to the decision biases, while the error consistency quantified based on Cohen's Kappa captures both shared behavior biases and consistency in their underlying decision variables. This point is further demonstrated using a simple simulation (see Fig. 5e). Here, we add behavioral biases to the two observers by shifting the decision criteria consistently, so that they both prefer one category over another. Naturally, as the bias increases, the accuracy drops. Cohen's Kappa strongly reflects these decision biases, whereas DVC remains consistent (Fig.5g). Furthermore, DVC can effectively decouple accuracy v.s. trial-by-trial consistency of the decision variables [20].

# 5   Discussion

We have developed a method, DVC, to quantify the consistency of two neural representations. The focus on task-relevant features makes DVC different from other popular approaches such as RSA[15] or linear regression [5]. Two representations may have high DVC yet low consistency according to RSA, or vice versa. For behavioral metrics that aim to characterize trial-by-trial consistency, one challenge has been to decouple task performance and trial-by-trial consistency. DVC provides a principled way to do so. For future work, it would be interesting to combine the analysis of DVC at the behavioral level [20] and the neural representation level to dissect the contribution of consistency of DVs and the shared biases of the observers. It would also be interesting to systematically compare DVC to other proposed similarity measures [15; 5; 19; 17][6]. Applying the DVC method, we find that there are surprising negative correlations between (i) the classification performance of the deep networks trained on ImageNet-1k and (ii) the consistency with the neural representation in V4/IT. Furthermore, training the deep network adversarially or using rich datasets seems to evoke decrease, rather than increase of DVCs. While it is unclear how to close the gap between the image-by-image consistency of deep networks to that of the brain, the following directions might be promising: (i) training networks using datasets that better resemble the stimulus statistics that drives the evolution of the primate visual system [31]; (ii) develop training procedures that better capture the stimulus noise and internal noise of the brain [51], as well as low level properties of the visual system (e.g., optics and foveation).

**Limitations** First, our results are limited by the number of monkey subjects in the datasets and the number of neurons recorded simultaneously from V4 and IT. Future larger neural datasets would allow for more accurate estimates of DVCs. Second, despite various adjustments in dimensionality reduction and DV decoding that we have experimented with, there may be factors that we have not taken into account that limit the scope and applicability of the results. For example, It is possible that the high-dimensionality of the feature space of some models affected the estimation of the DV. However, the DVCs of these models with the monkeys are not systematically lower, thus it is unlikely that they are underestimated. Third, while monkeys provide access to neural recordings, the objects shown in the experiments might not have the same behavioral relevance for them as they do for humans. Thus, caution should be taken when attempting to generalize the result to humans.

---

[5]See Appendix D.3 for results on the 'edges' data using a logistic regression classifier.

[6]See Appendix E for some preliminary results on comparison to RSA.

**Acknowledgments** We thank Nikolaus Kriegeskorte for helpful discussions. This research was supported by the National Institute Of Neurological Disorders And Stroke of the National Institutes of Health (Award Number R01NS133924), the NIH BRAIN Initiative and National Institute On Drug Abuse (Award Number R01DA060742), and a Sloan Research Fellowship (to X.-X. Wei). The content is solely the responsibility of the authors and does not necessarily represent the official views of the funding agencies.

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

# A  Method details

## A.1  Split normalization recovers true DVC

Assume the noisy DVs:
$$\mathrm{DV}_A = s_A + \varepsilon_A \qquad \mathrm{DV}_B = s_B + \varepsilon_B$$

Assume mean-centered and all signal-noise and noise-noise covariances vanish:
$$\mathbb{E}[s_A] = \mathbb{E}[s_B] = \mathbb{E}[\varepsilon_A] = \mathbb{E}[\varepsilon_B] = 0$$
$$\mathrm{Cov}(s_A, \varepsilon_A) = \mathrm{Cov}(s_A, \varepsilon_B) = \mathrm{Cov}(s_B, \varepsilon_A) = \mathrm{Cov}(s_B, \varepsilon_B) = \mathrm{Cov}(\varepsilon_B, \varepsilon_B) = 0$$

Note:
$$\mathrm{Var}(s_A) = \sigma_A^2, \quad \mathrm{Var}(s_B) = \sigma_B^2, \quad \mathrm{Cov}(s_A, s_B) = \rho_{\mathrm{true}}\sigma_A\sigma_B$$
$$\mathrm{Var}(\varepsilon_A) = \sigma_{\varepsilon_A}^2, \quad \mathrm{Var}(\varepsilon_B) = \sigma_{\varepsilon_B}^2$$

Then:
$$\mathrm{Cov}(\mathrm{DV}_A, \mathrm{DV}_B) = \mathrm{Cov}(s_A, s_B) = \rho_{\mathrm{true}}\sigma_A\sigma_B$$

$$\mathrm{Var}(\mathrm{DV}_A) = \sigma_A^2 + \sigma_{\varepsilon_A}^2, \quad \mathrm{Var}(\mathrm{DV}_B) = \sigma_B^2 + \sigma_{\varepsilon_B}^2$$

So the observed correlation is:
$$\rho_{\mathrm{obs}} = \rho_{\mathrm{true}} \cdot \frac{\sigma_A\sigma_B}{\sqrt{(\sigma_A^2 + \sigma_{\varepsilon_A}^2)(\sigma_B^2 + \sigma_{\varepsilon_B}^2)}}$$

Now split both DVs:
$$\mathrm{DV}_{A1} = s_A + \varepsilon_{A1}, \quad \mathrm{DV}_{A2} = s_A + \varepsilon_{A2}, \qquad \mathrm{DV}_{B1} = s_B + \varepsilon_{B1}, \quad \mathrm{DV}_{B2} = s_B + \varepsilon_{B2}$$

Assuming independent, identically distributed splits, and zero-mean and zero-covariances as before:
$$\mathrm{Var}(\mathrm{DV}_{A1}) = \sigma_A^2 + \sigma_{\varepsilon_A}^2, \quad \mathrm{Cov}(\mathrm{DV}_{A1}, \mathrm{DV}_{A2}) = \sigma_A^2$$

So the within-observer reliability is:
$$\rho(\mathrm{DV}_{A1}, \mathrm{DV}_{A2}) = \frac{\sigma_A^2}{\sigma_A^2 + \sigma_{\varepsilon_A}^2}$$

Likewise for B:
$$\rho(\mathrm{DV}_{B1}, \mathrm{DV}_{B2}) = \frac{\sigma_B^2}{\sigma_B^2 + \sigma_{\varepsilon_B}^2}$$

And the cross-observer split correlations are:
$$\rho(\mathrm{DV}_{Ai}, \mathrm{DV}_{Bj}) = \rho_{\mathrm{true}} \cdot \frac{\sigma_A\sigma_B}{\sqrt{(\sigma_A^2 + \sigma_{\varepsilon_A}^2)(\sigma_B^2 + \sigma_{\varepsilon_B}^2)}}$$

where $i, j = (1, 2)$.

Taking the geometric mean of the cross-observer split correlation gives a better estimate of $\rho_{\mathrm{obs}}$:
$$r_{\mathrm{cross}} = [\rho(\mathrm{DV}_{A1}, \mathrm{DV}_{B1}) \cdot \rho(\mathrm{DV}_{A1}, \mathrm{DV}_{B2}) \cdot \rho(\mathrm{DV}_{A2}, \mathrm{DV}_{B1}) \cdot \rho(\mathrm{DV}_{A2}, \mathrm{DV}_{B2})]^{1/4} = \rho_{\mathrm{obs}},$$

While the geometric mean of the within-observer split correlation gives the normalization factor:
$$r_{\mathrm{self}} = [\rho(\mathrm{DV}_{A1}, \mathrm{DV}_{A2}) \cdot \rho(\mathrm{DV}_{B1}, \mathrm{DV}_{B2})]^{1/2} = \sqrt{\frac{\sigma_A^2}{\sigma_A^2 + \sigma_{\varepsilon_A}^2} \cdot \frac{\sigma_B^2}{\sigma_B^2 + \sigma_{\varepsilon_B}^2}} = \frac{\sigma_A\sigma_B}{\sqrt{(\sigma_A^2 + \sigma_{\varepsilon_A}^2)(\sigma_B^2 + \sigma_{\varepsilon_B}^2)}}$$

Then the noise-corrected correlation is:
$$\rho_{\mathrm{corrected}} = \frac{r_{\mathrm{cross}}}{r_{\mathrm{self}}} = \frac{\rho_{\mathrm{true}}\sigma_A\sigma_B / \sqrt{(\sigma_A^2 + \sigma_{\varepsilon_A}^2)(\sigma_B^2 + \sigma_{\varepsilon_B}^2)}}{\sigma_A\sigma_B / \sqrt{(\sigma_A^2 + \sigma_{\varepsilon_A}^2)(\sigma_B^2 + \sigma_{\varepsilon_B}^2)}} = \rho_{\mathrm{true}}$$

### A.2 Upper bound on Cohen's Kappa with respect to accuracy difference between the two observers

Let $I, J \in \{0, 1\}$ be the correctness indicators of two decision makers on the same $N$ items, with accuracies $p_i = \mathbb{P}(I = 1)$, $p_j = \mathbb{P}(J = 1)$, and $d = |p_i - p_j|$. Define

$$c_{\text{obs}} = \mathbb{P}(I = J), \qquad c_{\text{exp}} = p_i p_j + (1 - p_i)(1 - p_j), \qquad \kappa = \frac{c_{\text{obs}} - c_{\text{exp}}}{1 - c_{\text{exp}}} \quad (c_{\text{exp}} < 1).$$

$c_{\text{obs}}$ is maximized when $I, J$ maximally agree, and $p_i, p_j$ only differ by $d$. Hence $c_{\text{obs}} \leq 1 - d$, and

$$c_{\text{obs,max}} = 1 - d.$$

For $c_{\text{exp}}$, parametrize the accuracies by their mean $m = \frac{p_i + p_j}{2}$:

$$p_i = m + \tfrac{d}{2}, \qquad p_j = m - \tfrac{d}{2}, \qquad m \in \left[\tfrac{d}{2}, 1 - \tfrac{d}{2}\right].$$

Then

$$c_{\text{exp}}(m) = p_i p_j + (1 - p_i)(1 - p_j) = 1 - (p_i + p_j) + 2 p_i p_j = 2m^2 - 2m + 1 - \frac{d^2}{2},$$

the expression is minimized at $m^\star = \frac{1}{2}$, yielding

$$c_{\text{exp,min}} = \frac{1}{2} - \frac{d^2}{2}.$$

Since $\kappa = (c_{\text{obs}} - c_{\text{exp}})/(1 - c_{\text{exp}})$ is increasing in $c_{\text{obs}}$ and decreasing in $c_{\text{exp}}$ (for $c_{\text{exp}} < 1$), combining the results gives

$$\kappa \leq \frac{(1 - d) - \left(\frac{1}{2} - \frac{d^2}{2}\right)}{1 - \left(\frac{1}{2} - \frac{d^2}{2}\right)} = \frac{\frac{1}{2} - d + \frac{d^2}{2}}{\frac{1}{2} + \frac{d^2}{2}} = \frac{(1 - d)^2}{1 + d^2}.$$

Tightness is achieved by $m^\star = \frac{1}{2}$ (i.e., $p_i = \frac{1+d}{2}$, $p_j = \frac{1-d}{2}$) and maximum agreement between $I, J$, which realizes $c_{\text{obs}} = 1 - d$ and $c_{\text{exp}} = \frac{1}{2} - \frac{d^2}{2}$.

### A.3 Simulation demonstrates the relationship between bias, accuracy and Cohen's Kappa

In order to demonstrate that this decision bias could influence Cohen's Kappa, we did a simple simulation. Suppose that there are 10 classes with 100 samples each. The observers output a vector which corresponds to the 10 classes. An unbiased perfect observer outputs 'DV (decision variable)' 1 for the corresponding class and 0 for all other classes (a one-hot vector). For realism, as observers make mistakes, we simply added gaussian noise to the DV output, which results in both lower Cohen's Kappa and lower DVC. To model a biased imperfect observer, a bias is applied after DV, which is the same for all samples in the same class (e.g. 0.1 for the first class, 0.2 for the second class) etc. Varying bias levels is achieved by scaling the bias added to the output. The final output is one-hot + noise + bias.

Here, Cohen's Kappa is directly inflated by the shared bias between two observers. On the other hand, because the bias does not affect the underlying DVs, the pre-normalization DVC is unaffected by the addition of bias. However, DVC does become systematically lower when the DVs are dominated by noise. Therefore, Cohen's Kappa and DVC are distinct in that the former cares about the decision criterion and the latter do not. They can be seen as complementary in certain scenarios. The simple simulation also hints at the relationship between accuracy, bias, and Cohen's Kappa. We continue this discussion in section D, where we reiterate that Cohen's Kappa is intimately linked to accuracy.

# B   Model and dataset details

## B.1   Model performances and choices of the penultimate layers

Table A.1: Models Trained on ImageNet-1k

| Model Name | Top-1 Acc | Top-5 Acc | Model Family | Layer |
|---|---|---|---|---|
| alexnet | 56.522 | 79.066 | AlexNet | classifier[-3] |
| vgg11_bn | 70.37 | 89.81 | VGG | classifier[-3] |
| vgg13_bn | 71.586 | 90.374 | VGG | classifier[-3] |
| vgg16_bn | 73.36 | 91.516 | VGG | classifier[-3] |
| vgg19_bn | 74.218 | 91.842 | VGG | classifier[-3] |
| squeezenet1_0 | 58.092 | 80.42 | SqueezeNet | features[-1] |
| squeezenet1_1 | 58.178 | 80.624 | SqueezeNet | features[-1] |
| densenet121 | 74.434 | 91.972 | DenseNet | features.norm5 |
| densenet169 | 75.6 | 92.806 | DenseNet | features.norm5 |
| densenet201 | 76.896 | 93.37 | DenseNet | features.norm5 |
| inception_v3 | 77.294 | 93.45 | Inception | avgpool |
| resnet18 | 69.758 | 89.078 | ResNet | avgpool |
| resnet34 | 73.314 | 91.42 | ResNet | avgpool |
| resnet50 | 76.13 | 92.862 | ResNet | avgpool |
| resnet101 | 77.374 | 93.546 | ResNet | avgpool |
| resnet152 | 78.312 | 94.046 | ResNet | avgpool |
| shufflenet_v2_x0_5 | 60.552 | 81.746 | ShuffleNet | conv5 |
| mobilenet_v2 | 71.878 | 90.286 | MobileNet | classifier[0] |
| resnext50_32x4d | 77.618 | 93.698 | ResNet | avgpool |
| resnext101_32x8d | 79.312 | 94.526 | ResNet | avgpool |
| wide_resnet50_2 | 78.468 | 94.086 | ResNet | avgpool |
| wide_resnet101_2 | 78.848 | 94.284 | ResNet | avgpool |
| mnasnet0_5 | 67.734 | 87.49 | MNASNet | classifier[0] |
| mnasnet1_0 | 73.456 | 91.51 | MNASNet | classifier[0] |
| googlenet | 69.778 | 89.53 | GoogLeNet | avgpool |
| convnext_base | 84.062 | 96.87 | ConvNeXt | avgpool |
| convnext_tiny | 82.52 | 96.146 | ConvNeXt | avgpool |
| convnext_small | 83.616 | 96.65 | ConvNeXt | avgpool |
| convnext_large | 84.414 | 96.976 | ConvNeXt | avgpool |
| efficientnet_b0 | 77.692 | 93.532 | EfficientNet | avgpool |
| efficientnet_b4 | 83.384 | 96.594 | EfficientNet | avgpool |
| efficientnet_b7 | 84.122 | 96.908 | EfficientNet | avgpool |
| efficientnet_v2_s | 84.228 | 96.878 | EfficientNet | avgpool |
| efficientnet_v2_m | 85.112 | 97.156 | EfficientNet | avgpool |
| regnet_y_8gf | 82.828 | 96.33 | RegNet | avgpool |
| regnet_y_16gf | 82.886 | 96.328 | RegNet | avgpool |
| swin_b | 83.582 | 96.64 | Swin | avgpool |
| swin_v2_b | 84.112 | 96.864 | Swin | avgpool |
| swin_v2_s | 83.712 | 96.816 | Swin | avgpool |
| swin_v2_t | 82.072 | 96.132 | Swin | avgpool |
| vit_b_16 | 81.072 | 95.318 | ViT | encoder.ln |
| vit_b_32 | 75.912 | 92.466 | ViT | encoder.ln |
| vit_l_16 | 79.662 | 94.638 | ViT | encoder.ln |

Table A.2: Robust Models

| Model ID | Architecture | Clean Acc | Robust Acc | Layer |
|---|---|---|---|---|
| Liu2023Comprehensive_Swin-L | Swin-L | 78.92 | 59.56 | norm |
| Liu2023Comprehensive_ConvNeXt-L | ConvNeXt-L | 78.02 | 58.48 | norm |
| Liu2023Comprehensive_Swin-B | Swin-B | 76.16 | 56.16 | norm |
| Singh2023Revisiting_ViT-B-ConvStem | ViT-B + ConvStem | 76.3 | 54.66 | norm |
| Peng2023Robust | WideResNet-101-2 | 73.44 | 48.94 | avgpool |
| Chen2024Data_WRN_50_2 | WideResNet-50-2 | 68.76 | 40.6 | avgpool |
| Salman2020Do_50_2 | WideResNet-50-2 | 68.46 | 38.14 | avgpool |
| Salman2020Do_R50 | ResNet-50 | 64.02 | 34.96 | avgpool |
| Engstrom2019Robustness | ResNet-50 | 62.56 | 29.22 | avgpool |
| Salman2020Do_R18 | ResNet-18 | 52.92 | 25.32 | avgpool |

Table A.3: Data-rich Models

| Model Name | Architecture | Top-1 Acc | Training | Layer |
|---|---|---|---|---|
| resnetv2_50x1_bitm | ResNetV2 (BiT-M) | 80 | ImageNet-21k | norm |
| resnetv2_50x3_bitm | ResNetV2 (BiT-M) | 82.6 | ImageNet-21k | norm |
| resnetv2_101x1_bitm | ResNetV2 (BiT-M) | 81.5 | ImageNet-21k | norm |
| resnetv2_101x3_bitm | ResNetV2 (BiT-M) | 84 | ImageNet-21k | norm |
| resnetv2_152x2_bitm | ResNetV2 (BiT-M) | 83.7 | ImageNet-21k | norm |
| resnetv2_152x4_bitm | ResNetV2 (BiT-M) | 84.3 | ImageNet-21k | norm |
| tf_efficientnet_l2.ns_jft_in1k_475 | EfficientNet-L2 | 88.4 | Noisy Student + JFT | pool |
| regnet_y_16gf_swag_e2e | RegNetY-16GF | 86 | hashtAGs | avgpool |
| regnet_y_32gf_swag_e2e | RegNetY-32GF | 86.8 | hashtAGs | avgpool |
| regnet_y_128gf_swag_e2e | RegNetY-128GF | 88.2 | hashtAGs | avgpool |
| vit_b_16_swag_e2e | ViT-B/16 | 85.3 | hashtAGs | encoder.ln |
| vit_l_16_swag_e2e | ViT-L/16 | 88.1 | hashtAGs | encoder.ln |
| CLIP | ViT-B/32 | NA | Image-text pairs | NA |

While we do not have a strict criterion for selecting which models to test, we do follow certain principles. First of all, we try to cover a diverse set of model architectures and span the range of model accuracy, which is why we included older models with mediocre performances. Secondly, we try to include models that other studies have previously examined, so it is easier to compare our study with the previous studies. We did exclude some models due to time limits. We intend to examine an even more comprehensive set of models in future work.

## B.2 Licenses for Third-Party Assets

The models used in this study were sourced from RobustBench, Torchvision, and Timm (PyTorch Image Models). We use these pretrained models as a cohort to study representational similarity, without referring to their individual implementation details.

We make use of publicly available datasets and pretrained models in accordance with their respective licenses:

- Brain-Score/Vision dataset (Majaj et al., 2015) were used solely for non-commercial academic research. We follow the terms of use as outlined.
- Models from Torchvision are provided under the BSD 3-Clause License.
- Robustbench models are used under the MIT License.
- Timm models are used under the Apache License 2.0.

# C   Implementation and verification

For detailed implementation used in analysis, refer to `https://github.com/wei-bbc-lab/DVC`.

## C.1   General information on the DVC framework

The DVC method is computationally efficient and stable, as dimensionality reduction is applied before attempting to decode the DV using LDA. All experiments were performed on Intel(R) Core i7-14700K CPU without resorting to GPU usage. Computing DVC between a pair of models takes 30 seconds on average, with the total compute rounding to 30 hours.

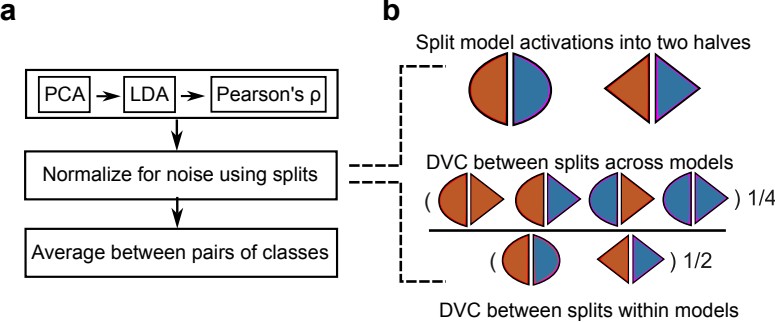

Figure A.1: **Implementation of the DVC framework**. (a) The diagram of our anlaysis pipeline. To increase the accuracy of the decision variable inferred in the regime of huge dimensionality and few samples, we first reduce the dimensionality of the neural representation (or hidden layer representations in neural networks) before applying the optimal linear classifier to infer the decision variables. (b) To correct for the under-estimation of the magnitude of the inferred decision variable due to noise, we develop a normalization procedure based on estimating the effective noise from two splits of the data. See text for more details.

## C.2 Verification of the robustness of DVC results

While the guiding principle behind DVC is general and intuitive, specific implementation choices carry implications on the numerical stability and robustness to different data distributions. We thus experimented with different algorithms and hyperparameter choices and found that they do not affect the main conclusions drawn in this study. Specifically, we want to verify if the choice of PC dimensions might change the conclusions in this study. First we note that with 25 PC dimensions (for each split), all the monkey recordings and model representations achieve high binary linear seprability (Fig. A.2a), and that 8-way logistic regression accuracy plateaus early on (Fig. A.2b). In addition, the main result is robust with varying PC dimensions (Fig. A.3a,b).

Other choices also do not affect the results. Pearson's correlation is easily biased by extreme values. Although the DVs appear normally distributed, we substituted Pearson's correlation with Spearman rank correlation and found that the trend persists (Fig. A.3c). In addition, we tested shrink regularization for LDA, which could enhance stability of the procedure, and find that the result is robust to the choice of LDA solver (Fig. A.3d).

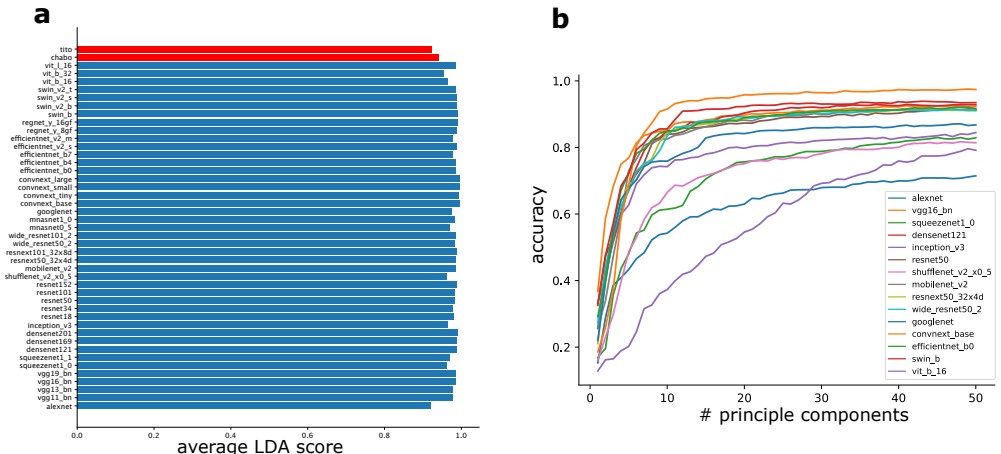

Figure A.2: Dimensionality reduction retains task-relevant information. (a) LDA score of all monkeys (red) and networks are high. (b) As the dimensionality increases, decoding performance using logistic regression plateaus.

## C.3 Consistency of the DVC results across different linear decoders

The choice of using LDA as a decoder is not coincidental. Under the assumption of Gaussian distribution and in the setting of pairwise classification, a linear decoder provides the best classifier. We calculated DVC between the two monkey subjects using logistic regression and linear support vector machine (SVM). We find that the DVC is highly consistence across different linear decoders. In addition, across repeats (random split normalization), DVC estimation varied little. On the other hand, DVC values calculated using nonlinear decoders such as kernel SVM and multi-layer perceptron are significantly lower, suggesting overfitting (Fig. A.3e).

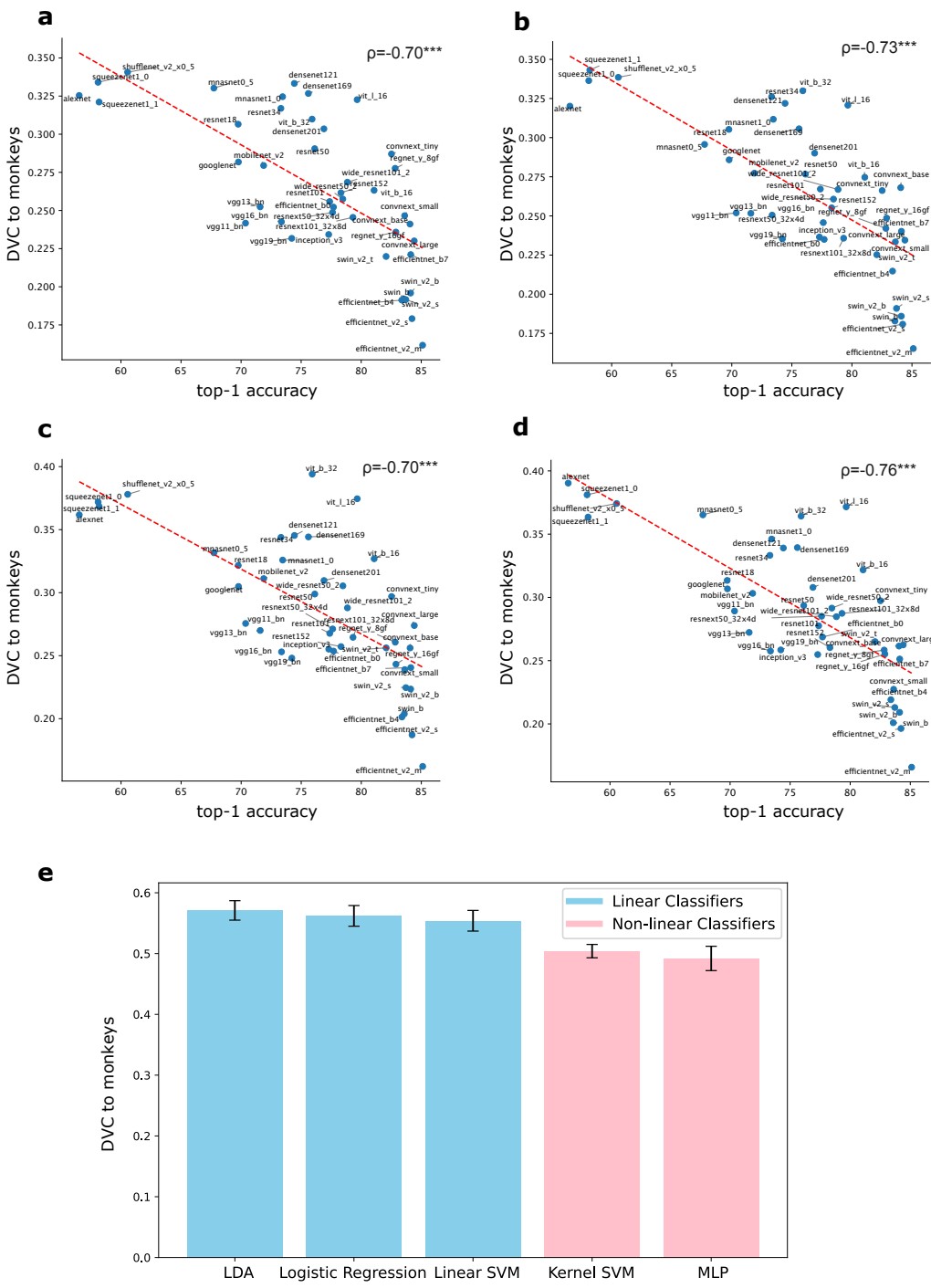

Figure A.3: The main results are robust to particular choices in the implementation. We tested the result in Fig. 2c where networks with better ImageNet-1k performance are less alignment under different conditions when (a) using the top 10 principal components (PCs) instead of 25. (b) Using the top 50 PCs instead of 25. (c) Using Spearman's rank correlation instead of Pearson's correlation. (d) Using an eigen SVD solver with shrinkage instead of a SVD solver for LDA. (e) Measure of monkey-monkey DVC is consistent across linear decoders and across random splits. Error bar indicates the standard deviation.

### C.4 DVC results are consistent in another neural datasets

To alleviate concern that the dataset from which we derived the main results contains only two monkeys, we verified the result on an additional dataset from Bashivan et al [52]. The dataset contains three monkey subjects viewing natural and synthesized images. Among the three subjects, only one ('monkey M') has sufficient neuron count for this test (n = 168). The neurons are pooled across four recording sessions. We used data from monkey M viewing naturalistic images and calculated its DVC against neural networks viewing the same images. The main results are also significant in this case (Pearson's correlation = -0.84, p = 0.87e-05, Fig.A.4a).

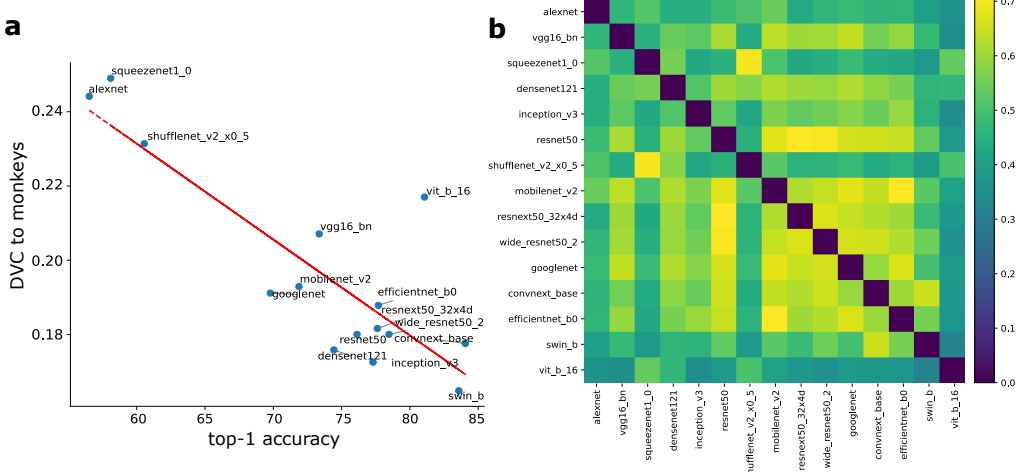

Figure A.4: The main result is confirmed in an additional dataset from [52]. (a) DVC of monkey M against representative models. Again, these is a strong negative correlation between the top-1 accuracy of the deep networks and the DVC to the monkey brain. (b) DVC between representative models.

### C.5 DVC between representations and raw image pixels

We calculated the DVC between the V4/IT representations and model hidden representations to flattened image pixels and found that the DVC between the two monkeys and raw image pixels are 0.153 and 0.195, respectively. The DVC between model hidden representations and network DVC are generally lower but still substantial (Fig. A.5a)). The results suggest that part of the DVC results might come from remnants of linear separability in the image pixels, which is reduced by nonlinearity in the brain and models. Indeed, the LDA score for categorization using raw image pixels is 0.70, which is smaller than that of brain and model representations (>0.90) but still very high. We plotted the LDA axis for classifying 'chair' and 'car' in image pixel space and see clear silouettes of the objects (Fig. A.5b), suggesting that high DVC to image pixels may be due to simplicity of generated image samples. It would be interesting to test the result on datasets with more complex and diverse images.

### C.6 Split normalization under uncommon conditions

The split normalization procedure could behave counterintuitively under extreme conditions. For example, it could result in a DVC value larger than 1 when the internal noise is high. None of the normalized correlations between different representations reached the cap. In addition, we took the absolute value before calculating the geometric means. When there is no correlation between two representations, this would bias the normalized DVC to a small positive value. None of the representations in this study fall in this range.

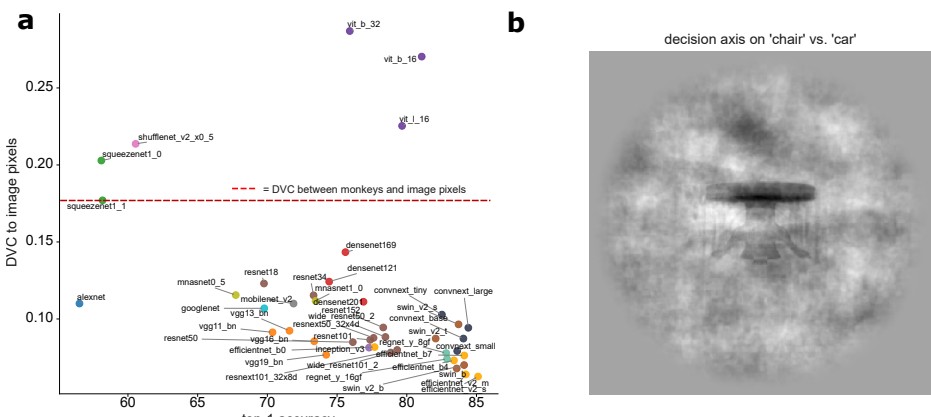

Figure A.5: DVC between the monkey and model representations and raw image pixels. (a) The DVC between models and raw image pixels are markedly lower than DVC between models and monkeys yet still substantial. (b) The LDA axis between categories 'chair' and 'car' in image pixel space.

# D  Additional results on the effect of behavioral decoding on Cohen's Kappa

Geirhos et al. [13] have discussed several caveats of applying Cohen's Kappa, including that Cohen's Kappa is bounded by error overlap expected by chance $c_{exp}$. We also explicitly showed that the accuracy difference $d = |p_i - p_j|$ provides an upper bound on Cohen's Kappa (see Section A.2). When the accuracy difference is high, one subject is often right, while the other is often wrong, then their behaviors cannot be consistent. From the original behavioral data, it is clear that human subjects perform with high accuracy but models perform poorly(Fig. A.6). While Geirhos et al used simulations to show that under the condition that the subjects act independently, accuracy is not necessarily correlated with model performance, Cohen's Kappa may still depend on accuracy under more general conditions.

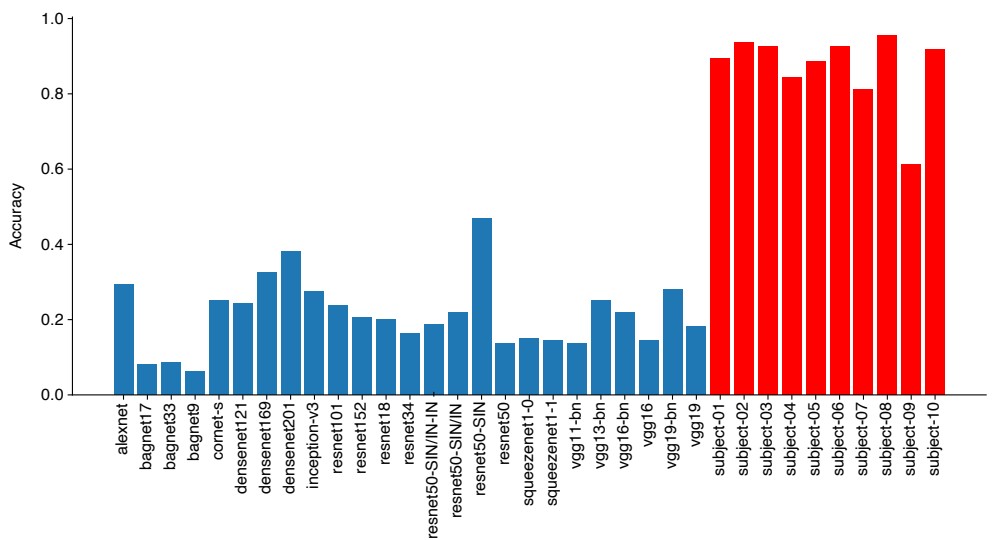

Figure A.6: Re-plotting of the data (based on "edge" stimuli from [13]. Red: accuracy of human subjects. Blue: accuracy of deep network models tested. Overall, the accuracy of the deep network models are much lower than that of human subjects.

We would like to verify whether the high model-model consistency reported in the original work might be inflated by the low-accuracy high-bias condition caused by the choice of decoder (directly aggregating the probabilities). To test this, we take the original stimuli provided by Geirhos et al., and calculate Cohen's Kappa between the models by (i) taking the average of the probability of the corresponding ImageNet-1k subclasses or (ii) training a 5-fold cross-validated logistic regression decoder on the representations in the penultimate layer. The result shows that compared to (i), approach (ii) achieves higher accuracy (Fig. A.7b,c), exhibits less bias towards certain categories (Fig. A.7d,e) and results in significantly lower model-model consistency as measured by Cohen's Kappa (Fig. A.7a).

While the shared behavioral bias that results from aggregating probabilities from the original ImageNet-1k classes is very interesting, it does make the human-model consistency and the model-model consistency a lot more ambiguous. Thus, it may be more appropriate to use a stronger decoder for classification or to use DVC when applicable.

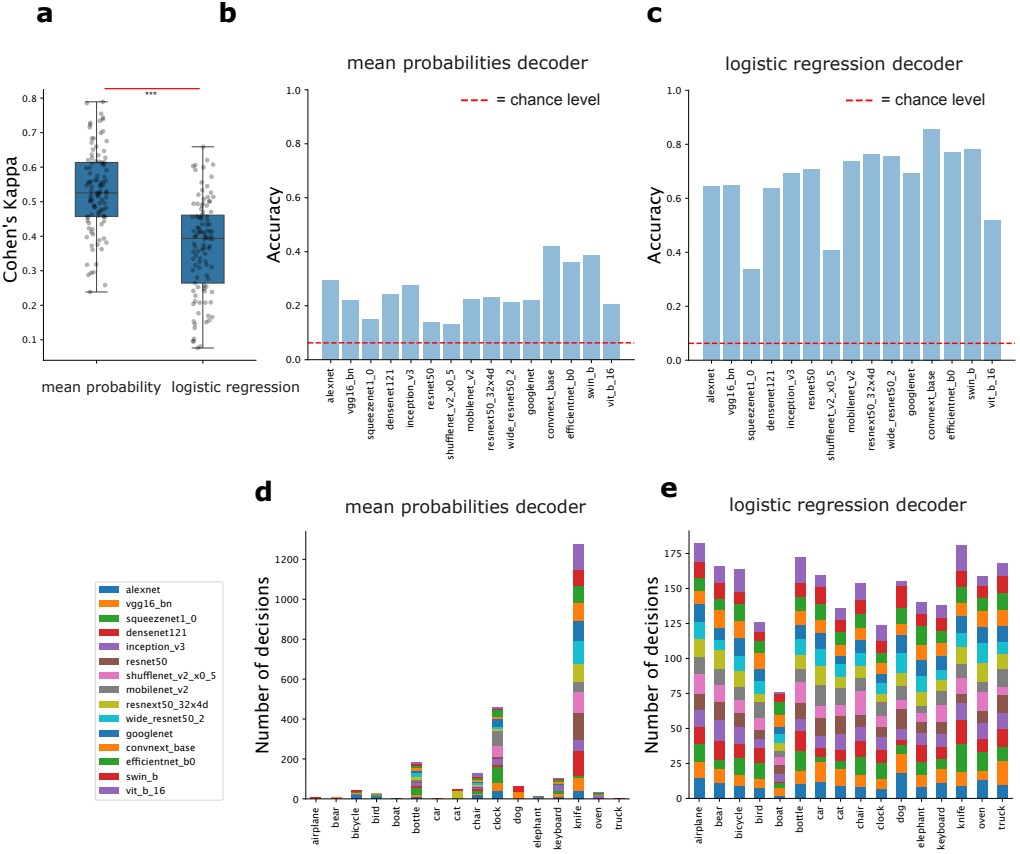

Figure A.7: Using logistic regression decoder results in higher accuracy, lower bias and lower Cohen's Kappa estimate compared to mean probability decoder, estimated on the 'edges' images. (a) Cohen's Kappa estimated using a mean probability decoder is significantly higher than that estimated by a logistic regression decoder. (b) Behavioral accuracy of mean probability decoder. (c) Behavioral accuracy of logistic regression decoder. (d) Choice histogram of the mean probabilities decoder. (e) Choice histogram of the logistic regression decoder.

# E    Comparison to Representational Similarity Analysis

Representational similarity analysis (RSA) measures the similarity between the general arrangement of categories (or samples) between two representations. Under appropriate conditions, it can be seen as equivalent to centered kernel alignment (CKA) and canonical correlation analysis (CCA) [53]. In the monkey dataset that we tested, RSA is positively correlated with DVC when measuring both model-monkey alignments and model-model alignments (see Fig. A.8a-d)).

While RSA may contain information about the correlations of task-related dimensions, they may be confounded by task-irrelevant correlations. We demonstrate this point using a simple simulation. Given the same baseline decision variables ('task-relevant correlation'), we introduced independent fluctuations ('noise') and shared fluctuations ('task-irrelevant correlation'). We find that while DVC is generally unaffected by the introduction of stronger task-irrelevant correlation (Fig. A.8e). In contrast, RSA quickly plateaus as task-irrelevant correlation overshadows task-relevant ones. This simulation demonstrates that RSA and DVC capture different aspects of the representation. The ability of DVC to focus on only task-relevant dimensions makes it suitable when applied to scenarios with high shared fluctuation based on common stimulus input, where only some dimensions encode task-relevant information.

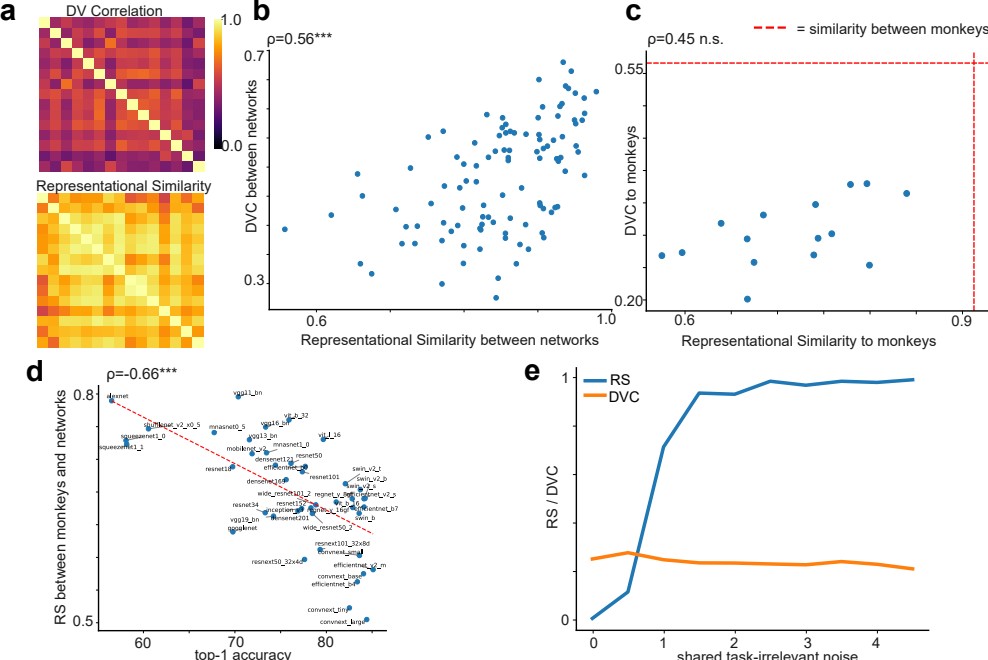

Figure A.8: **Comparison to Representational Similarity Analysis (RSA)**. (a) Heatmaps showing the DVC and representational similarity (RS) for pairs of representative models. (b) There is a strong positive correlation between DVC and RS on model-model consistency (evaluated on this dataset). (c) The correlation between category-level RSA. (d) Category-level RSA recreates the main result that alignment between model and brain declines with ImageNet top-1 accuracy. (e) Simulation results show that RSA is conflated with task-irrelevant correlations whereas DVC only cares about task-relevant ones.

## F   Broader Impacts

We expect DVC to be a broadly applicable approach to study the similarity of brain and neural network models. On the positive side, the method and results discussed in this paper could redirect community focus away from brute-force scaling and toward more targeted investigations into task-relevant representation alignment and model-brain convergence. It could in the long term lead to models that are more brain-like, thus greatly facilitating research in fields like neuroscience, cognitive science and AI interpretability and safety. However, while DVC offers a biologically grounded lens for comparing model and brain representations, promoting alignment with biological brains might inadvertently constrain models in certain domains where brain cognition is suboptimal.

