# OpenReview forum: "Quantifying Task-relevant Similarities in Representations Using Decision Variable Correlations"
_NeurIPS.cc/2025/Conference — NeurIPS 2025 poster_

### Official Review · Reviewer_4tHi · 2025-06-24

**Clarity:** 4
**Significance:** 3
**Originality:** 3
**Rating:** 5
**Confidence:** 4

**Summary:**

The paper proposes a novel method for measuring the similarity of decision strategies between two observers, as is frequently done between DNNs and primate brains in the field of NeuroAI. The method is called decision variable correlation (DVC), which aggregates the trial-level correlations between the decision variables of the observers, which is basically the distance a sample in embedding space has to the separatrix of a linear classifier.
Using this method, the authors find low between-group similarity between monkeys and models despite high within-group similarity. This low between-group similarity is not increased by increased classification performance on ImageNet, by more training data, or by higher adversarial robustness, which suggests profound representational differences between monkey brains and image classification models.

**Questions:**

- What is the effect of the split normalization (sec. 3.2 second paragraph / Appendix A) on results? How would results look like without this normalization? If the point of this step is to reduce noise, wouldn't it be better to split the images into different sets, rather than the neurons?
- What is the motivation behind checking all pairwise decision variables, rather than just looking at the class-vs-all-others separatrix, which would scale much nicer to large numbers of classes, like 1k ImageNet classes?
- What is the uncertainty in a DVC-value? You currently just report a point estimate without specifying a CI around these values. This could be achieved by bootstrapping, i.e. if a value is calculated over 1,000 images, draw 1,000 images from the set with replacement for N times and re-calculate DVC for each set, then report the range in which 95% of values end up.
- What is a model's self-DVC? For example, train a ResNet-50 from scratch on ImageNet with two different random seeds and evaluate their DVC on the validation set.
- Similarly, what is the DVC between two randomly initialized networks, is it really 0 as expected, and what is the variance?
- What is the classification accuracy that an optimal linear classifier on top of the monkey representatins achieves?
- (section 4.5, second paragraph) I'm surprised that logistic regression on top of network representations does not beat the aggregation strategy proposed by Geirhos et al,  shouldn't it be at least as good?

**Ethical Concerns:**

["NO or VERY MINOR ethics concerns only"]

**Final Justification:**

In my initial review, I had some concerns about the paper, mostly because there are already too many metrics of alignment rather than too few. There were also some choices that were not properly ablated. However, during the rebuttal period, the authors have addressed the specific concerns I had raised and clarified some details of the method. It seems that this approach is comparatively well-grounded in signal detection theory, and results seem to be robust. At this stage, I have no good arguments against accepting this paper, so vote to accept, which generally also seems to be the consensus among reviewers.

**Limitations:**

The authors have written a limitation-section, but I think missed one point which they maybe thought was obvious: The method is limited to datasets that have class labels, like ImageNet. This is not true for all image datasets, and other methods like RSA work independently of known class labels, hence I think it should be explicitly mentioned.

**Paper Formatting Concerns:**

I see no formatting concerns.

**Quality:**

3

**Strengths And Weaknesses:**

## Strengths
- The paper addresses a relevant problem and is fairly clearly written.
- The idea of only using those parts of the representation that are relevant for the task is valuable.
- The conducted analysis is very thorough, sufficiently many models were investigated and the selection of models is reasonable.

## Weaknesses
My main gripe with the paper is that they propose yet another method for measuring brain-model similarity, and immediately jump to deriving conclusions, instead of first sanity-checking the method itself. I propose a few of these sanity checks in the Questions-block.

- The description of the method lacks rigor, the authors never properly define what the decision variable is. I understood it to be the length of the projection onto the separatrix, but I'm not even sure if this is correct.
- I find the split-normalization highly dubious. The authors just assume that within the two randomly drawn subpopulations of neurons of a layer, the same signal is represented, plus different noise, which can essentially be averaged out. I don't share this intuition, because if the different subgroups represent different features, the signal may be different.
- The comparison to Cohen's Kappa (especially how figure 5e was obtained) is missing details, e.g. why does a low decision bias enforce a low kappa?
- I have some concerns about the dependence of DVC on the accuracy of the two classifiers, which are not addressed by the paper. I suspect that two classifiers with very different accuracy cannot have a high DVC, just like two classifiers with different accuracies cannot have a high Error Consistency (as measured by Cohen's kappa over correctness values). It does not surprise me that increased accuracy on ImageNet-1k does not yield increased DVC to monkeys, because I doubt that an optimal linear classifier on top of the monkey representations would perform very well. I fear that such a dependency on accuracy could also explain the findings in figure 2d, because (as seen in figure 2c) models from the same family tend to have similar accuracy. This is problematic, as it would mean that I gain no more knowledge from DVCs than I would trivially get from accuracy.

## Conclusion
Overall, I find the method interesting and could imagine accepting the paper, because it meaningfully adds to the discourse. But there are too many issues that make me feel uneasy about the method, so I don't trust the results yet. If the authors address these concerns and other reviewers agree, I'm open to increasing my score.

---

> ### Author Rebuttal · Authors · 2025-07-31
>
> Thank you for your very detailed feedback and questions on our work. For several of these points, we have now performed additional analysis to address them. We feel that by addressing these critiques, the paper has been greatly improved.
>
> (1) Explain the decision variable: Your intuition is not far off! The decision variable is a term derived from signal detection theory that is frequently adopted by behavioral psychologists as a model for decision making, which is why it is the perfect base ingredient for this method. It is a continuous variable that represents the sensory evidence of the observer. A threshold (the separatrix) is further used to make the decision. So here it is the projection onto the axis orthogonal to the separatrix.
>
> (2)Justify split normalization: Your question about if the split halves are truly two copies of the same information (decision variable) contaminated by independent noise is an interesting one.
> Following your suggestion, we have performed additional analysis by decoding the individual splits. Although there is no ground truth available for the decision variable, we find the decoding performance to be high for the individual splits. In addition the DVC estimates with random splits are very consistent across repeats (normalized DVC for monkey vs. monkey is 0.568 ± 0.014), showing that it is likely that splits contain the same copy of task-relevant information. This result is consistent with the general idea that information about the task is distributed in the network. The split normalization, as you would expect, rescues DVC relating to noisier observers, making the monkey-monkey, monkey-model DVC higher than would otherwise be. As the math (described in the Supplementary Information section A1) shows, under the correct (if somewhat restrictive) assumptions, the procedure recovers the ground truth DV.
>
> The splits are on the neurons instead of the images because we are trying to account for the overall noise in the representation instead of the image sampling noise. The sample size is large enough in this case (400 images in each category) that the latter is miniscule. However this could be useful in cases where the sample size is smaller.
>
> (3)The relationship between Kappa and bias: This is a good question. The details on the simulation that addresses the relation between Cohen’s Kappa and decision biases are provided in the Supplementary Materials Section A2. We apologize if the description has been unclear. The general intuition is that DVC is agnostic to the behavioral biases of the observers while Kappa is inflated when there is a shared bias between observers. Low bias does not enforce a low Kappa.
>
> (4)Accuracy of linear decoder on monkey recordings: Thank you for this suggestion. An optimal linear decoder on monkey V4/IT recordings actually performs very well (average pairwise LDA score 0.92 & 0.94 for the two monkeys respectively, which is at a similar level as the networks ~0.96) so it is very unlikely to be the cause of differences in DVC. The accuracy for the logistic regression classifier for all the classes (instead of binary classification) is 0.80 & 0.81, respectively. With cross validation it drops to 0.74 & 0.71. In addition, the decision variable is conceptually decoupled from accuracy, since it cares about the information used to make the decision, not the decision itself. Which makes it great for addressing behavioral differences by different observers. We will include this result into the revised version of the paper.
>
> (5)Pairwise or one vs. the others: Our choice of focusing on pairwise classification instead of one vs. the others classification is in part due to the fact that we use a smaller dataset. However, also consider that between two (e.g. gaussian) distributions with equal covariances, the best classifier is linear. But in the case of one vs. many gaussian distributions, it can be very hard, or even impossible, to find a good linear classifier depending on the arrangement of the classes, especially in settings where the effective signal dimension is low.
>
> (6)The uncertainty of DVC: DVC estimate with split is very robust with the amount of samples in this dataset. As previously mentioned, normalized DVC for monkey vs. monkey is 0.568 ± 0.014, where the uncertainty comes from different splits. The uncertainty of DVC with respect to sampling can be estimated using bootstrap procedure. Thank you for this suggestion.
>
> (7)DVC between networks initiated with different seeds: As established in studies such as Mehrer et al. 2020 (Nat Comm), networks with varying initiation can learn very different intermediate representations, therefore should be viewed as different models. To address your question, we tested DVC on Resnet50 variants trained using different initialization, augmentations, training recipes etc. Notably, DVC between models with different initialization but otherwise underwent the same training procedure is 0.80.
>
> (8)DVC results with randomly initialized networks: We have since examined DVC on randomly initialized networks. Under special circumstances where networks of the same family are initialized using the same seed, DVC reports could be quite high. However in general DVC between randomly initialized networks fall in the 0.10 - 0.20 range.
>
> Thank you again for taking time to carefully examine our manuscript. We hope our new analyses and the clarification above address your concerns. Please feel free to follow up with further inquiries.

---

> > ### Comment · Reviewer_4tHi · 2025-08-05
> >
> > Thank you for the extensive response. I am impressed by the breadth of additional experiments that have been conducted during the rebuttal phase and apologize for not being able to respond earlier.
> >
> > (1) Thank you for the explanation of the decision variable -- while I acknowledge that I could have been expected to know this, I still feel like a brief explanation would benefit the paper.
> >
> > (2) The results on the different splits are very interesting, I did not expect this result but greatly appreciate that you conducted the experiment. It has resolved my concerns about this point.
> >
> > (3) "Low bias does not enforce a low Kappa."
> > I am still confused about this point: I understood figure 5e to be implying this, as here, low decision bias leads to a small Cohen's Kappa. (By the way: I think a better term would be "Error Consistency", as Cohen's Kappa can in principle be calculated over the raw responses themselves, whereas for Error Consistency it is clear that the correctness values are used for the calculation. Or do you, unlike Geirhos et al, consider Cohen's Kappa directly over the responses?)
> >
> > (4) Again, thank you for conducting this experiment, I appreciate it and it alleviates my concerns. I have also convinced myself with a toy example in 2d that it is possible to have observers with differing accuracy but high DVC, I think. This is a nice property of the metric and an advantage over Error Consistency!
> >
> > (5) Thanks, convincing argument.
> >
> > (6 & 7) I admit that maybe these are not the best sanity checks, just the best ones I could think of -- I see a tendency in the field of people proposing new metrics and using them before sanity-checking them, which I think is a problem, as misleading results could proliferate before somebody revisits the metric and points out the limitations (reviewer 9A6o seems to agree with the sentiment that we have too many metrics rather than too few). But maybe this is just part of scientific progress and it seems like your metric is comparatively well-grounded in SDT.
> >
> > Overall, the rebuttal has positively affected my view of the paper and I will increase my rating to reflect this.

---

> > > ### Author Response · Authors · 2025-08-07
> > >
> > > Thank you for once again taking care to examine our arguments.
> > >
> > > DVC and bias: in signal detection theory the sensory evidence is first represented by decision variables. The observers then use criterions (which reflect the bias) to make decisions. Here the simulation mainly shows that Cohen's kappa is inflated by the bias of the observers. Two low bias observers can definitely have high Cohen's kappa, as long as their decisions are in agreement. The difference between DVC and Cohen's kappa is that DVC does not care about the bias of observers since bias is reflected by the criterions. We will revise this figure and the explanation to avoid confusions.
> > >
> > > Cohen's kappa vs. error consistency: Like you described, error consistency extends Cohen's kappa to mean correct / incorrect. In figure 5 we are still in the regime of binary classification, where the final reported value is the average pairwise Cohen's kappa, so Cohen's kappa and error consistency are interchangeable in this case. We will make this more clear in the manuscript.
> > >
> > > Sanity-checking: We want the readers to feel as we do that DVC is unique and grounded, much more than just another metric. The constant stream of constructive feedbacks has been extremely helpful. We hope the additional analysis will better illustrate the limitations and strengths of the method.

---

### Official Review · Reviewer_4fv6 · 2025-07-01

**Clarity:** 3
**Significance:** 3
**Originality:** 2
**Rating:** 5
**Confidence:** 4

**Summary:**

This paper proposes Decision Variable Correlation (DVC) as a new method to measure task-relevant similarity between neural representations in brains and deep networks.  DVC focuses on the trial-by-trial correlation of inferred decision variables, isolating task-relevant information while being robust to noise and decision bias. Applying DVC to monkey V4/IT and various deep networks, the authors find that model–model consistency is comparable to monkey–monkey, but model–monkey consistency is lower and decreases with higher ImageNet accuracy. Neither adversarial training nor larger datasets improve model–brain alignment, suggesting a divergence between artificial and biological representations.

**Questions:**

**Questions and Suggestions for the Authors**

1. **Comparison to RSA/CKA**: The abstract and introduction motivate DVC by highlighting the limitations of standard representational similarity methods (e.g., RSA, CKA). However, the paper does not empirically compare DVC to these methods. Can the authors include such comparisons—ideally using the same model and brain representations? These metrics can also be applied to the output (decision) layer, allowing for a fairer comparison to DVC, which is decoded via LDA from the same representations.

2. **Clarifying Novelty**: Several of the qualitative findings (e.g., that adversarial training or dataset scale does not improve brain alignment) have been previously reported using other similarity metrics. Can the authors clarify what new insights DVC provides beyond replicating existing conclusions?

**Ethical Concerns:**

["NO or VERY MINOR ethics concerns only"]

**Final Justification:**

I still think that more experiments could be helpful to make the use case of this method more distinguishable compared to the existing methods. However, based on clarifications provided by authors, and their promises to add clarifications to the text, I am now more confident to recommend this work for publication.

**Limitations:**

yes

**Paper Formatting Concerns:**

Looks fine.

**Quality:**

3

**Strengths And Weaknesses:**

**Strengths**

- **Clarity and Motivation:** The paper is clearly written and well-motivated. It introduces a conceptually principled method (DVC) that isolates task-relevant similarity, avoiding confounds like accuracy and decision bias. The derivation from signal detection theory is elegant and intuitive.
- **Sound Methodology:** The method is grounded in theory, and the implementation accounts for practical issues such as noise correction and high-dimensional instability. The authors take care to validate that DVC behaves sensibly under these conditions.
- **Conservative Claims:** The paper avoids overclaiming and discusses its limitations transparently, including dataset size and generalizability beyond monkeys.

**Weaknesses**

- **No Empirical Comparison to RSA/CKA:** While the paper argues that DVC improves on representational similarity methods like RSA, it does not empirically compare DVC to RSA, CKA, or CCA. This weakens the case for DVC as a superior alternative, especially since RSA-based methods have also reported similar qualitative conclusions (e.g., limited brain-model alignment).
- **Overlap with Prior Findings:** Many key takeaways—such as the failure of adversarial training or larger datasets to improve alignment—have been reported previously using other similarity metrics. Without a side-by-side comparison, it remains unclear how much new insight DVC offers.

---

> ### Author Rebuttal · Authors · 2025-07-31
>
> Thank you for your constructive feedback. The concerns you have are very representative and we have worked on addressing them.
>
> (1)Comparisons to other methods: Thank you for raising this great point. RSA, CKA and CCA are all useful correlation-based metrics and equivalent under certain conditions. We have compared these metrics in this dataset and find that DVC is indeed positively correlated with all the aforementioned metrics, albeit to different degrees. Of course, DVC is a supervised method that utilizes the labels to discern task-relevant dimensions from task-irrelevant ones, which distinguishes it from other correlation-based methods. The RSA between two simulated representations may be arbitrarily inflated while DVC remains the same. The fact that they are positively correlated, and the degree to which they are correlated, could help us pick apart how much of the variance in the representation is useful for task-solving. We will incorporate the new results on the comparison to relevant other methods in the final version of the paper. Thank you for your valuable suggestion which helped us improve the paper.
>
> (2)Clarifying novelty: While the main contribution of the paper is the DVC method, we hope that we could use a principled, task-driven approach to examine the topics where other methods have been giving contradictory accounts. Take for example, the relationship between brain alignment and Imagenet accuracy, papers have reported increasing, saturating, peaking then dropping etc. While DVC reports show that the alignment has been dropping consistently since early days, with the exception of some noteworthy outliers. The finding that alignment between brains are on par with the alignment between models also contradicts previous results. We think additional novelty of these findings lies in the method used to derive them and the principles behind the method, making the findings more grounded than previous reports. We agree with the reviewer that these points regarding the novelty should be made more clear in the paper. We will revise the paper to make these points more salient.

---

> > ### Comment · Reviewer_4fv6 · 2025-08-03
> >
> > Thank you for the clarifications. I have increased my score and look forward to seeing the additions the authors intend to make if the paper is accepted for publication.

---

### Official Review · Reviewer_9A6o · 2025-07-02

**Clarity:** 3
**Significance:** 2
**Originality:** 3
**Rating:** 2
**Confidence:** 4

**Summary:**

This paper introduces a new representational similarity measure for stimulus-driven neural activity. The method is based on a multidimensional generalization of signal detection theory. Neural responses to different stimuli with a known class structure are projected onto the LDA (linear discriminant analysis) axis. The Pearson correlation between the projections of these stimuli for two observers (modulo a noise correction factor) is the proposed similarity score between the neural representations of the corresponding observers. Some advantages of the proposed measure over purely behavioral measures (Cohen’s kappa) are discussed (e.g. insensitivity to decision biases, dependence on internal (neural) representations) and a seeming discrepancy with an earlier study using a purely behavioral measure (Cohen’s kappa) is resolved by taking into account the decision biases affecting Cohen’s kappa.

**Questions:**

Please address the issues raised under the "weaknesses" section above. In particular, the authors need to do a much better job motivating their approach: why exactly it is needed and what it offers over the multitude of current methods.

**Ethical Concerns:**

["NO or VERY MINOR ethics concerns only"]

**Final Justification:**

The proposed method has a rather severe limitation: it only applies to classification tasks. This limitation does not hold for alternative methods like RSA or various flavors of regression: these are universally applicable methods for any type of task or stimuli. **The authors have not shown any deficiencies in these alternative methods that their own method is addressing.**

Moreover, the authors misleadingly pit their method against Cohen's kappa, which is a well-established measure of **decision correlations** for different observers. The paper and the rebuttal persistently (and again misleadingly to my mind) refer to the method as measuring the correlation between "decision strategies of two observers" or variations thereof, but actually, unlike Cohen's kappa, **the proposed method has nothing to do with the actual decisions of the observers** (only with the "decisions" of an optimal linear classifier), so these methods are not directly comparable. The proposed method is rather more accurately thought of as an alternative to representational similarity measures like RSA or regression, but as mentioned before, the authors have not shown any deficiencies in these particular methods; on the contrary, the proposed method has a severe limitation compared to these alternative representational similarity measures.

So, in my opinion, the proposed method is unmotivated and unnecessary. I'm sorry for being a bit harsh here, but my honest judgment is that it is simply cluttering an already cluttered literature on representational similarity.

**Limitations:**

I think the issues raised above are limitations of the work that aren't adequately addressed by the authors.

**Paper Formatting Concerns:**

None.

**Quality:**

2

**Strengths And Weaknesses:**

**Strengths:**

I think this is a technically solid paper overall. The experimental demonstration in section 4.5 that Cohen’s kappa used in earlier analyses can be affected by decision biases is a nice contribution. It’s also nice to see error bars in a NeurIPS submission for a change.

**Weaknesses:**

I guess my main concern about this work is that I’m not sure if we really needed another representational similarity measure. In the first two sections of the paper, the authors complain about the multiplicity of representational similarity measures and how they can sometimes give seemingly contradictory results. But they are making this problem worse by introducing (without any good justification) yet another similarity measure that is not going to always agree with previous measures.

What exactly is the problem with RSA or regression-based metrics (like Brain-Score), for example, that this new method is addressing? In section 3, the authors write (lines 96-98):
> ... in contrast to methods for analyzing the similarity of two neural representations (such as representational similarity analysis), our methods focus on the dimensions that are relevant for a behavioral task and is invariant to variability along other task-irrelevant dimensions.

I’m not sure if this is a win for the proposed approach. The proposed approach can’t be applied to cases where there is no obvious class structure in the stimuli or where the class structure is irrelevant (*e.g.* passive viewing of natural videos or images).

Even in cases where there is a class structure and it is relevant to the task, somewhat counterintuitively, the proposed method actually doesn’t say anything about the observer’s actual behavior in the task (it only says something about the behavior of an ideal linear classifier based on the neural responses, which may be completely different from the observer’s own behavior). Whatever its problems, Cohen’s kappa at least doesn’t suffer from this particular issue: it says something about the observer’s actual behavior.

So, it appears that the proposed method is intended more as an alternative to neural similarity measures, such as RSA or regression-based metrics, as opposed to behavioral similarity measures like Cohen's kappa, but the authors haven't really shown any particular problems with these neural similarity measures that their approach is addressing. The problem they identify only affects Cohen's kappa. So, we're left wondering what exactly is the pressing need or the real motivation behind this new neural representational similarity measure.

If I’m interested in quantifying the similarity between neural responses and model responses, I can use RSA or a regression-based metric. These are general methods I can use for any type of stimuli or task. Their application is not limited to classification tasks. If I don’t have access to neural responses or if I just want to quantify the behavioral similarity between an observer and a model, I can use kappa or some bias corrected version of it. I haven’t really read anything in this paper that made me question these choices and convinced me that a new similarity measure was desperately needed.

---

> ### Author Rebuttal · Authors · 2025-07-31
>
> Thank you for your thoughtful review on our work. The main criticism is why our method is needed given there are already several prior methods to modeling the alignment of neural/behavioral responses across different observers/models. This is an important point that requires further clarification.
>
> (1)Reason to adopt DVC over other methods: As a representational similarity measure, DVC is a supervised method that utilizes labels for task-specific comparisons, which distinguishes it from the methods that focus on general geometry. In addition, it is less biased and more principled. We think it should be able supersede previous methods where the conditions for its application are met.
>
> DVC can be also applied strictly to behavior such as comparing models to behavior, comparing two behaving animals, or the same behaving animal on multiple occasions. In all cases, it provided an unbiased principled measure of decision correlations. For more on how DVC may be inferred from behavior, see Sebastian & Geisler, 2018.
>
> (2)Scope of application: Biological vision systems evolve to perform two kinds of specific tasks that are relevant for survival: categorization tasks, and estimation tasks. DVC can be measured for both kinds of tasks. We are interested in understanding biological vision. Most of the rigorous studies of biological vision have been some form of categorization or estimation task. A rigorous unbiased measurement technique should find plenty of applications.
>
> It is true that there are tasks that the DVC methods would not apply, but we have deliberatively developed our method to study a specific kind of question, in particular, how well the decision strategies of two observers are correlated on a trial-by-trial basis in classification tasks. This is a unique feature of our method that distinguishes our work from other methods, such as RSA which quantifies the similarity of the geometry of two representations.
>
> We appreciate your thoughtful comments and will revise the paper to incorporate the key points above.

---

> > ### Comment · Reviewer_9A6o · 2025-08-05
> >
> > Descriptions of the method such as "how well the **decision strategies** of two observers are correlated" and "unbiased principled measure of **decision correlations**" seem misleading. As it is currently described in the paper, the method only considers the "decisions" of an optimal linear classifier (LDA), not the actual decisions of the observers.

---

> > > ### Author Response · Authors · 2025-08-07
> > >
> > > Thank you for raising this concern. In the conception of DVC, it was used by the authors to study behavioral alignment between two observers. In this work, the DV is inferred from representations rather than behavior. Consistent with studies in neuroscience and cognitive science, decision variables extracted by LDA reflect the decisions of a Bayes-optimal observer under the assumption of linearly separable classes with Gaussian distributions and equal covariances.  We agree that we should take care so readers do not confuse them with decisions from actual behavioral data, and will resort to using language such as 'inferred decision strategies' in the manuscript and in future communications to avoid ambiguity.

---

### Official Review · Reviewer_4PbU · 2025-07-03

**Clarity:** 4
**Significance:** 3
**Originality:** 4
**Rating:** 5
**Confidence:** 4

**Summary:**

1. New approach (Decision Variable Correlation) to characterize the similarity of the decision strategies of two modalities referred to as observers in this paper
2. Model-monkey similarity is consistently lower than monkey-monkey or model-model relationship
3. Adversarial training , Improved accuracy on Imagenet or pretraining on larger dataset does not increase model-monkey similarity

The study demonstrates divergenece between task relevant representations in monkey V4/IT and models trained on image classification tasks.

**Questions:**

Did you consider applying DVC to opensource human neuroimaging datasets?

**Ethical Concerns:**

["NO or VERY MINOR ethics concerns only"]

**Final Justification:**

The authors addressed all my concerns. I will maintain the original rating. The authors may consider THINGS dataset for furture experiments

**Limitations:**

yes

**Quality:**

4

**Strengths And Weaknesses:**

**Strength**

1. DVC quantifies task relevant similarities and is insensitive to the observer’s biases and independent of behavioral accuracy. It’s a nice bridge between methods purely based on behavior and methods comparing neural representations and it focuses only on those dimensions which are relvant for the behavioral task
2. Comprehensive Evaluations:
    1. Large set of models to identify correlation between imagenet accuracy and model-monkey DVC
    2. Robustly trained models vs. representative models
    3. Dependency of monkey-model DVC on scale of data used for model training
3. Interesting findings:
    1. Data rich and robustly trained model show lower DVC than representative models. The finding is surprising and reveals difference between how monkeys and models solve classification tasks
    2. Comparison with Cohen’s Kappa measure shows that DVC is insensitive to decision biases.
4. Clear description of methods, results and figures

**Weakness**

1. A theoretical/data-driven experiment using some simulated data to demonstrate the key differences between different measures would have revealed the pros and cons of different behavioral and representational similarity methods.

---

> ### Author Rebuttal · Authors · 2025-07-31
>
> Thank you for your positive evaluation of our paper and for providing great ideas to help further strengthen it. We address your suggestions below.
>
> (1)Simulation to show difference between methods
> This is a great suggestion. The difference between unsupervised shape metrics such as RSA and DVC is that DVC focuses on the task dimension and ignores similarities that are irrelevant to the task. To demonstrate this, we simulated two observers with representations that consisted of 1D decision variable, individual noise and shared noise. As shared noise increases, RSA quickly increases and saturates towards 1, while DVC remains stable. We feel this is a great point to add to the paper because it makes the motivation of our method more clear. We will incorporate this point to the revised version of the paper.
>
> (2)DVC to neuroimaging datasets: The existing neuroimaging datasets (e.g. Horikawa & Kamitani, 2017) typically contain hundreds or thousands of image categories, but only a handful of exemplar images in each category, which is the opposite of what we are looking for in this case (more images per category). We were looking at the popular Natural Scenes Dataset (NSD Dataset) which uses COCO images, which has no fixed number of images per category, and each image has multiple labels. This makes the NSD dataset not ideal for our purpose.
> If the reviewer can think of any publicly available datasets that would be appropriate for the application of our approach, please share them with us. It would be greatly appreciated and would be eager to test these out.

---

### Official Review · Reviewer_7UD3 · 2025-07-03

**Clarity:** 4
**Significance:** 3
**Originality:** 3
**Rating:** 4
**Confidence:** 4

**Summary:**

This paper introduces a novel method, Decision Variable Correlation (DVC), to measure the similarity between the decision strategies of different observers (e.g., brains and deep neural networks). Grounded in Signal Detection Theory, DVC quantifies the trial-by-trial consistency of task-relevant representations by correlating the decision variables decoded from two observers for a binary classification task. Applying this method to compare monkey V4/IT neural recordings with a wide range of deep learning models, the authors uncover several surprising findings. They report that model-monkey similarity is consistently lower than monkey-monkey similarity and, counter-intuitively, decreases as models' ImageNet performance increases. Furthermore, neither adversarial training nor pre-training on larger datasets appears to close this gap. These results lead the authors to suggest a fundamental divergence between the representations learned by current models and those in the primate visual cortex.

**Questions:**

**Missing Citations:** The findings that adversarial training increases model-model similarity but decreases model-brain alignment are consistent with other recent work. Have the authors considered citing papers like Subramanian et al. (2023) and Linsley et al. (2023) to better contextualize their results and show how they fit into a broader emerging consensus?

**Generalizability of Claims:** The claim of a "fundamental divergence" is quite strong. How confident are the authors in this conclusion given that the neural data comes from only two monkeys? Could the observed model-monkey gap be partially explained by inter-animal variability that a larger sample of primate data might clarify?

**Robustness of the DVC Method:** The DVC calculation relies on an optimal linear classifier (LDA) to define the decision axis. How sensitive are the reported DVC results to this specific methodological choice? Have the authors explored whether using a non-linear decoder or a different linear classifier would significantly alter the observed correlations and the paper's main conclusions?

**Ethical Concerns:**

["NO or VERY MINOR ethics concerns only"]

**Limitations:**

yes

**Quality:**

3

**Strengths And Weaknesses:**

### Strengths

**Novel and Principled Method:** The paper introduces DVC, a methodologically sound approach for comparing representations that is well-grounded in Signal Detection Theory. The careful consideration of potential confounds, such as the use of PCA to stabilize the decoder and the development of a split-half procedure to correct for measurement noise, demonstrates rigor (Section 3).

**Comprehensive Experimental Scope:** The study's conclusions are supported by an extensive evaluation across a diverse set of 65 models, including standard ImageNet-trained networks, adversarially robust models, and models trained on web-scale datasets (Section 4). This breadth makes the findings less likely to be an artifact of a specific architecture or training scheme.

**Significant and Counter-intuitive Findings:** The paper's central results challenge common assumptions in the field. The strong negative correlation between ImageNet accuracy and DVC to monkey V4/IT (Fig. 2c) and the observation that adversarial training actually reduces model-monkey similarity (Fig. 3a) are important, surprising findings that should stimulate further research.

**Strong Baselines and Contextualization:** The use of monkey-monkey DVC provides a clear biological benchmark, giving crucial context to the model-monkey similarity scores. Furthermore, the direct comparison to Cohen's Kappa, a related behavioral metric, helps to situate DVC within the literature and highlights its specific advantages, such as its insensitivity to decision bias (Section 4.5, Fig. 5e).

### Weaknesses and Possible Improvements
**Insufficient Contextualization with Recent Literature:** The paper's findings, particularly regarding adversarial training, align with a growing body of work suggesting that standard robustness metrics do not guarantee better alignment with biological vision. Citing recent, highly relevant papers (e.g., Subramanian et al., 2023; Linsley et al., 2023) would strengthen the authors' claims and better situate their contribution within the current scientific discourse.

**Limited Neural Dataset:** The strong claim of a "fundamental divergence" between models and the primate brain is based on neural recordings from only two macaque monkeys. While the authors acknowledge this in the limitations section, the forcefulness of their main claims could be tempered by this constraint. The potential for inter-subject variability in the primate data might be a larger factor than is currently accounted for.

**Sensitivity to Decoder Choice:** The entire DVC framework relies on using an optimal linear classifier (LDA) to find the "decision axis" for each observer. The paper does not explore how sensitive the final DVC values are to this specific choice. It is possible that a different decoder (e.g., a non-linear one, or even a different linear method) could yield different correlation patterns.

**References:**

1. Linsley, D., Feng, P., Boissin, T., Ashok, A. K., Fel, T., Olaiya, S., & Serre, T. (2023). Adversarial alignment: Breaking the trade-off between the strength of an attack and its relevance to human perception. arXiv preprint arXiv:2306.03229.

2. Subramanian, A., Sizikova, E., Majaj, N., & Pelli, D. (2023). Spatial-frequency channels, shape bias, and adversarial robustness. Advances in neural information processing systems, 36, 4137-4149.

---

> ### Author Rebuttal · Authors · 2025-07-31
>
> Thank you for your encouraging comments and constructive feedback. We find the criticisms to be very on point and have worked on addressing them with additional data.
>
> (1)Missing citations: Thank you for giving context to some of the findings. Subramanian et al. discovered that artificial neural networks use wider channels than human observers and robust networks use even wider channels. Linsley et al. show that harmonized networks are attacked on features that are more detectable and human-aligned. In future versions of the paper, we will provide the context in the vein of “There has been an emerging consensus that even though adversarial training increased model robustness, these robust networks do not use human-like features unless explicitly aligned”.
>
> (2)Generalizability/limited neural dataset:
> In the current version, we only reported data from Majaj et al., 2015, which is one of the best datasets that are available to us when considering the neurons recorded, trials repeated and data format.
>
> To address the reviewer’s concern, we have performed additional analysis and examined  more datasets. In particular, we have analyzed the dataset collected in Bashivan et al., 2019. This dataset contains only V4 neurons and a smaller number of recorded neurons – only one of the monkeys (monkey m) had sufficient neurons. As a result, the monkey-monkey DVC and monkey-model DVC concerning the monkey subjects with only ~20 neurons are lower due to insufficient sample size, but the relative scale of the DVC remains consistent. In addition, the observation that DVC drops with increasing ImageNet accuracy holds for monkey m (rho=-0.76, p=9.89e-04). We will edit the manuscripts so the words reflect the scope of implications accurately. Specifically, instead of “fundamental divergence”, we will use “substantial divergence”
>
> (3)Robustness to decoder choice:
> This is a good question. To address the reviewer’s concern, we have now additionally tested the DVC with other linear decoders such as logistic regression and linear SVM and found that the estimate remained stable (for DVC between monkeys: LDA 0.567 ± 0.014, logistic regression 0.562 ± 0.017, linear SVM 0.554 ± 0.017).
>
> Regarding nonlinear decoder, in the current work, we deliberately chose to focus on linear decoding because we are interested in understanding the trial-by-trial correlation of the linearly decodable information for classification tasks. The approach can be incorporated to deal with nonlinear decoder, however, the results obtained would be more difficult to interpret. For reference, using SVM with RBF kernel or MLP with ReLU activation seems to result in lower DVC across the board.
>
> We will incorporate these new results in the final version of the paper. Thank you again for your thoughtful and constructive advice which further strengthens the paper. We hope the additional results we provided above help mitigate your concerns.

---

### Note · Authors · 2025-08-14

Dear AC & Reviewers,

During the discussion period we received many constructive comments from the reviewers. We have, to the best of our abilities, performed additional analysis to address the technical concerns. Some reviewers suggested that the paper could benefit from extra explanations and motivations. We have provided our arguments and will improve upon the current manuscript using fortified language. The main points are summarized below:

Added analysis: 1) We tested DVC on an additional dataset and find that the main conclusions hold. 2) We tested DVC on alternative decoders and find that the estimate is consistent. 3) We find that DVC estimates under random splits are stable. 4) The linear decodability of the monkey V4/IT recordings is very high. 5) DVC between networks trained using different initialization is high. 6) DVC between randomly initialized networks are generally low but occasionally high.

Added motivation / explanations: 1) Unlike commonly used shape metrics such as RSA, CKA and CCA, DVC is unaffected by non-task related shared variance between two observers. We demonstrate this intuition using a simulation. 2) Unlike Cohen's kappa (error consistency), DVC is unaffected by bias exhibited by the observers. 3) We emphasized the novelty of the method and the results that contradict previous accounts. Further, we will better contextualize our result with new and relevant findings such as Subramanian et al., 2023 and Linsley et al., 2023. 4) We will provide a better explanation of concepts relating to signal detection theory. 5) We will add explanations to justify certain choices in the method such as linear decoders and pairwise classification. 6) We will adjust wordings to better reflect the scope and implications of the findings and to avoid confusion.

---

### Decision · Program_Chairs · 2025-09-17

**Decision:**

Accept (poster)

**Comment:**

This work proposes a representational similarity metric that is grounded in signal detection theory. They evaluate their metric on monkey V4/IT recordings and find the model-monkey similarity decreases with the model's performance on a ImageNet classification task, which is surprising given that other metrics (namely, linear regression) suggest the opposite.

Four of the reviewers recommended acceptance, while Reviewer 9A6o recommended rejection. Reviewer 9A6o's main concerns are that (1) the method is limited to classification tasks, and (2) the interpretation of their metric as a behavioral readout is misleading. During the discussion period, two of the other reviewers commented that they did not find (1) to be a significant limitation and that (2) could be addressed by the authors in their revision.

Given this, I recommend acceptance and urge the authors to clarify the issues raised by the reviewers.